# Massively Multimodal Foundation Models: A Framework for Capturing Interactions with Specialized Mixture-of-Experts

**Xing Han**
Johns Hopkins University
Baltimore, MD 21218, USA
xhan56@jhu.edu

**Hsing-Huan Chung & Joydeep Ghosh**
University of Texas at Austin
Austin, TX 78712, USA
{hhchung,jghosh}@utexas.edu

**Paul Pu Liang**[*]
Massachusetts Institute of Technology
Cambridge, MA 02139, USA
ppliang@mit.edu

**Suchi Saria**[*]
Johns Hopkins University
Baltimore, MD 21218, USA
ssaria1@jhu.edu

## Abstract

Modern applications increasingly involve many heterogeneous input streams, such as clinical sensors, wearable device data, imaging, and text, each with distinct measurement models, sampling rates, and noise characteristics. We define this as *massively multimodal* setting, where each sensor constitutes a separate modality. As modality counts grow, capturing their complex, time-varying interactions such as delayed physiological cascades between sensors, has becomes essential yet challenging. Mixture-of-Experts (MoE) architectures are naturally suited for this setting since their sparse routing mechanism enables efficient scaling across many modalities. However, existing MoE architectures route tokens based on similarity alone, overlooking the rich temporal dependencies across modalities: this prevents the model from capturing delayed cross-modal effects, leading to suboptimal expert specialization and reduced accuracy. We propose a framework that explicitly quantifies temporal dependencies between modality pairs across multiple discrete time intervals, defined as delays between an event in one input stream and its manifested effect in another, and uses these to guide MoE routing. A interaction-aware router dispatches tokens to specialized experts based on interaction type. This principled routing enables experts to learn generalizable interaction-processing skills. Experiments across healthcare, activity recognition, and affective computing benchmarks demonstrate substantial performance gains and interpretable routing patterns aligned with domain knowledge.

## 1 Introduction

Multimodal learning has traditionally focused on integrating two or three canonical modalities such as text, image, and audio (Baltrušaitis et al., 2018; Liang et al., 2024b). Yet real-world applications increasingly involve massively multimodal data: dozens to hundreds of heterogeneous input streams, each with distinct measurement models, sampling rates, noise characteristics, and temporal dynamics (Liang et al., 2021; 2022; Soenksen et al., 2022). In healthcare alone, a single patient generates continuous signals from heart rate monitors, pulse oximeters, blood pressure cuffs, ECGs, respiratory sensors, and laboratory analyzers, alongside imaging and clinical notes (Johnson et al., 2023). Each stream constitutes a distinct modality requiring different processing strategies (Soenksen et al., 2022). As modality counts grow, the space of cross-modal interactions explodes: some pairs provide redundant information, others capture unique signals, and still others exhibit synergistic patterns that emerge only through joint analysis (Williams & Beer, 2010; Liang et al., 2023). These dependencies often unfold over time with characteristic delays (Weissman et al., 2012;

---
*Equal Advising.

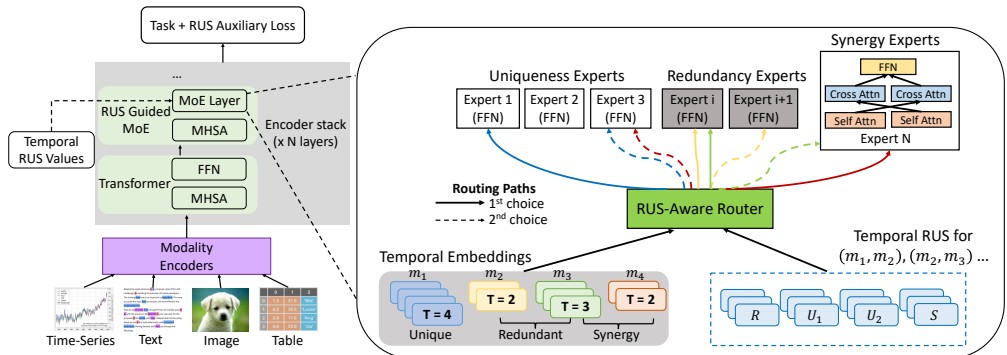

Figure 1: Overview of MERGE. The left panel illustrates the overall architecture, where multi-modal inputs are processed through $N$ stacked encoder layers composed of alternating Transformer and MoE blocks. The core innovation of MERGE lies in the MoE layers, detailed on the right. The temporal RUS-aware router is the essential part, which leverages temporal multimodal interactions to guide the routing of token embeddings across different time lags. The router determines, based on interaction dynamics, which modality pairs should (or should not) be routed to the same expert, thereby enabling more principled and interpretable expert specialization. As an example, $m_2$ (yellow) and $m_3$ (green) exhibit high redundancy according to their temporal RUS values; therefore, the router is more likely to assign them to the same expert (yellow and green arrows).

Varley, 2023), a predictable time intervals between a cause in one modality and its observable effect in another, further complicating the modeling challenge. In response, MoE models (Shazeer et al., 2017; Guo et al., 2025; Jiang et al., 2024) offer a natural foundation for this setting. It provides a principled way to allocate computation to modality-specific experts: LIMoE (Mustafa et al., 2022) designed multimodal MoE that processes both images and text through shared sparse MoEs with contrastive learning, demonstrating modality specialization through entropy regularization. Fuse-MoE (Han et al., 2024) addressed the "FlexiModal" setting with irregularly sampled and missing modalities, introducing a Laplace gating function with theoretical convergence guarantees superior to softmax routing in multimodal applications. Building on this, Flex-MoE (Yun et al., 2024) proposed a missing modality bank and dual-router design to handle arbitrary modality availability. Hierarchical MoE (Nguyen et al., 2024) demonstrated that Laplace gating at two hierarchical levels eliminates undesirable parameter interactions, accelerating expert convergence in multimodal tasks.

Despite the successful application of MoE in multimodal problems, most existing MoE architectures rely on routers that consider only the similarity between input tokens and experts; such routers only operate on static modalities (Riquelme et al., 2021; Fedus et al., 2022), or static features from temporal modalities (Xie et al., 2025; Zhu et al., 2025). However, in real-world applications, using multimodal interaction at a single time point often fails to capture the full landscape of cross-modal relationships, as delayed effects between variables or modalities are common (Zhang et al., 2025; Zhou et al., 2025). For example, when understanding human communication, a brief eyebrow raise followed 200–400 ms later by rising intonation and a half-smile turns the same words into sarcasm; in medical diagnosis, a slow overnight drift in $SpO_2$ and respiratory rate, followed hours later by fever and elevated lactate, flags early sepsis. This raises an essential question: *Can we incorporate temporal multimodal interaction to guide MoE training and inference?* To capture such temporal interactions, a successful MoE model should be able to: (1) employ a well-defined quantitative measure of multimodal interaction that accounts for time-delayed interactions between modalities, and (2) incorporate an enhanced MoE architecture capable of leveraging this information during training. With these components, the model can become more proactive and context-aware, leading to improved performance by better understanding complex, time-evolving multimodal processes.

Motivated by this challenge, we propose **M**assively-multimodal **E**xpert **R**outing for **G**eneralized **E**xchange (MERGE), a novel framework that leverages temporal multimodal interactions to guide the routing process of MoE. The overall framework is illustrated in Figure 1. We begin by introducing a method to explicitly compute temporal multimodal interactions (**R**edundancy, **U**niqueness, **S**ynergy, or RUS) between input modalities over time with respect to the target outcome. To handle high-dimensional and temporal data, we design a scalable approach based on multi-scale BATCH

estimator to compute multi-step temporal interactions efficiently. The resulting estimates are then used to guide the MoE training process. Specifically, we design an interaction-aware router that incorporates the context of temporal RUS sequences and dynamically routes tokens to experts based on these temporal interactions. This routing mechanism is further reinforced using auxiliary loss functions during training. The core of MERGE consists of two components: (1) computing temporal RUS (section 3.1), and (2) using the resulting RUS sequences to guide MoE training (section 3.2). We demonstrate the effectiveness of MERGE in two ways: first, by analyzing the insights provided by temporal RUS and showing that they capture meaningful and application-relevant patterns; and second, by illustrating that the learned routing patterns align with our expectations and lead to significant performance improvements across a diverse set of multimodal tasks.

## 2 RELATED WORKS

**Multimodal Interactions** define the degrees of commonality between modalities and the ways they combine to provide new information for a task (Liang et al., 2024b). A core problem lies in understanding the nature of how modalities interact and modeling these interactions using data-driven methods. The study of multimodal interactions has involved semantic definitions based on research in multimedia (Marsh & Domas White, 2003), verbal and nonverbal communication (Partan & Marler, 2005; Flom & Bahrick, 2007; Ruiz et al., 2006), social interactions (Mai et al., 2019; Jung et al., 2018), and instruction tuning (Shan et al., 2025). These have also inspired statistical methods to quantify multimodal interactions from unimodal predictions (Mazzetto et al., 2021), trained model weights and activations (Sorokina et al., 2008; Tsang et al., 2018; 2020; Hessel & Lee, 2020), feature selection (Ittner et al., 2021; Yu & Liu, 2003; 2004; Auffarth et al., 2010), and information theory (Williams & Beer, 2010; Bertschinger et al., 2014). For information theory-based methods, recent studies have investigated extensions to continuous (Pakman et al., 2021; Ehrlich et al., 2024) or Gaussian (Venkatesh et al., 2023) distributions, many modality decomposition (Varley, 2024), sample-level quantifications (Yang et al., 2025), and enable accurate estimation from large-scale multimodal datasets (Liang et al., 2023; 2024a). Recently, Varley (2023) introduced a framework for modeling multimodal interactions over time. While insightful, the approach has limitations: it does not naturally extend to continuous variables, and its lattice-based structure does not scale beyond small systems, limiting its practicality for large-scale multimodal applications.

**PID and Multimodal Interaction MoEs.** Partial Information Decomposition (PID) (Williams & Beer, 2010; Bertschinger et al., 2014) provides a principled framework for quantifying multimodal interactions by decomposing the total information into redundant ($R$), unique ($U$), and synergistic ($S$) components. Redundancy captures the shared information across modalities, uniqueness measures modality-specific contributions, and synergy reflects information that emerges only when modalities are combined. To the best of our knowledge, PID has not yet been exploited to improve multimodal MoEs. In contrast, existing multimodal MoE approaches typically rely on heuristic expert assignments (Shazeer et al., 2017; He et al., 2021; Han et al., 2022; Akbarian et al., 2024), which often result in opaque and difficult-to-interpret routing decisions (Liu et al., 2024; Shen et al., 2024). Multimodal interaction provides a natural and principled way to enhance expert specialization. For example, Yu et al. (2023) introduced the Mixture of Multimodal Interaction Experts, where specific experts are assigned to process predetermined types of modality interactions. Recently, Xin et al. (2025) extended this idea by incorporating interaction-type categorization into the MoE training process and expanding the framework to support more than two modalities. However, these designs make the number of experts tightly coupled to the number of modalities, which limits scalability. A more natural and flexible solution is to use multimodal interactions to guide expert routing, enabling the model to dynamically determine which modalities should (or should not) be processed together. Moreover, both works rely on a binary label agreement method based on unimodal classifiers to approximate the PID-type multimodal interactions, making the approach heavily dependent on the performance of the individual classifiers. Additionally, this method captures only static interactions, overlooking the temporal and continuous nature of real-world multimodal data.

## 3 RUS-GUIDED MOE FOR MASSIVELY MULTIMODAL LEARNING

We introduce MERGE, our framework for guiding MoE with temporal multimodal interactions in massively multimodal settings. The approach consists of two parts: (1) capturing temporal multimodal interactions, and (2) leveraging these quantified interactions to inform the training of MoE.

### 3.1 Capturing Temporal Multimodal Interactions

We first present our methodology for capturing temporal multimodal interactions. We detail the procedure for computing temporal RUS across different settings, describe how to efficiently estimate these interactions, and provide potential insights that can be derived from the temporal RUS patterns.

**Formulation of temporal RUS.** Capturing information interactions over time remains an important yet underexplored problem. *Our goal is to characterize how past and present values interact across time for a task.* For example, in the medical domain, one may wish to understand how the interactions of past treatments influence a patient's health outcomes; in activity recognition, it is crucial to capture how past motions of individual body parts (e.g., arms and legs) contribute to predicting future overall motion. However, the existing PID framework (Eq. 15–17 in Appendix A) is computed using standard mutual information, which only captures static interactions and cannot be directly extended to the temporal setting. To this end, we build on directed information (Weissman et al., 2012), which enables scalable modeling of temporal interactions while maintaining alignment with PID. Directed information respects temporal relationships by considering information flow from past to present, which enables PID analysis across multiple time lags, providing insights into both short-term and long-term influences. We begin by defining multi-source directed information, which quantifies the flow of information from multiple input modalities to the target variable over time.

**Definition 1** (Multi-Source Directed Information). *Given multiple source processes $X_1^{i-1} = (X_{1,1}, X_{1,2}, \ldots, X_{1,i-1})$, $X_2^{i-1} = (X_{2,1}, X_{2,2}, \ldots, X_{2,i-1})$ and the target process $Y^{i-1} = (Y_1, Y_2, \ldots, Y_{i-1})$, the directed information from $(X_1, X_2)$ to $Y$ over $n$ steps with time lag $\tau$ is:*

$$\text{DI}(\tau) = I(X_1^{n-\tau}, X_2^{n-\tau} \rightarrow Y^n) = \sum_{t=\tau+1}^{n} I(Y_t; X_{1,t-\tau}, X_{2,t-\tau}|Y^{t-1}). \tag{1}$$

As shown in Figure 2, at each time lag $\tau$, $\text{DI}(\tau)$ can be decomposed into four components based on the conditional joint distribution $P_\tau(x_1, x_2, y) = P(X_1^{n-\tau} = x_1, X_2^{n-\tau} = x_2, Y^n = y|Y^{n-1})$, where $\text{DI}(\tau) = R(\tau) + U_1(\tau) + U_2(\tau) + S(\tau)$: each of its component quantifies interaction at $\tau$:

$$R(\tau) = \min_{Q_\tau \in \Delta_\tau} I_{Q_\tau}(Y^n; X_1^{n-\tau}) + I_{Q_\tau}(Y^n; X_2^{n-\tau}) - I_{Q_\tau}(Y^n; X_1^{n-\tau}, X_2^{n-\tau}), \tag{2}$$

$$U_1(\tau) = I_{Q_\tau^*}(Y^n; X_1^{n-\tau}) - R(\tau), \quad U_2(\tau) = I_{Q_\tau^*}(Y^n; X_2^{n-\tau}) - R(\tau), \tag{3}$$

$$S(\tau) = I_{P_\tau}(Y^n; X_1^{n-\tau}, X_2^{n-\tau}) - I_{Q_\tau^*}(Y^n; X_1^{n-\tau}, X_2^{n-\tau}), \tag{4}$$

where $\Delta$ represents the probability simplex and $\Delta_\tau$ is the set of marginal-matching distributions defined as $\Delta_\tau := \{Q_\tau \in \Delta : Q_\tau(x_i, y) = P_\tau(x_i, y), \forall y \in Y, x_i \in X_i, i \in \{1, 2\}\}$; it characterizes the set of joint distributions $Q_\tau(x_1, x_2, y)$ that preserve the time-specific bivariate marginals $P_\tau(x_1, y)$ and $P_\tau(x_2, y)$ while allowing the coupling between $X_1$ and $X_2$ to vary; $Q_\tau^* = \arg\min_{Q_\tau \in \Delta_\tau} I_{Q_\tau}(X_1^{n-\tau}; X_2^{n-\tau}|Y^n)$ is the optimal distribution that has *eliminated the synergistic information* between $X_1$ and $X_2$. After further simplifying the formulation, we obtain

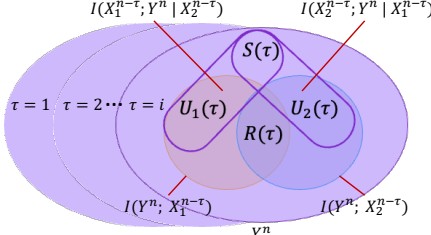

Figure 2: Decomposed directed information components across time lag $\tau$.

$$R(\tau) = \max_{Q_\tau \in \Delta_\tau} I_{Q_\tau}(X_1^{n-\tau}; X_2^{n-\tau}; Y^n), \tag{5}$$

$$U_1(\tau) = \min_{Q_\tau \in \Delta_\tau} I_{Q_\tau}(X_1^{n-\tau}; Y^n|X_2^{n-\tau}), \quad U_2(\tau) = \min_{Q_\tau \in \Delta_\tau} I_{Q_\tau}(X_2^{n-\tau}; Y^n|X_1^{n-\tau}), \tag{6}$$

$$S(\tau) = I_{P_\tau}(X_1^{n-\tau}, X_2^{n-\tau}; Y^n) - \min_{Q_\tau \in \Delta_\tau} I_{Q_\tau}(X_1^{n-\tau}, X_2^{n-\tau}; Y^n). \tag{7}$$

Note that, $X_1$ and $X_2$ do not necessarily have the same time lag $\tau$, and exhaustively enumerating all pairwise combinations of $\tau$ across modalities would further increase memory costs during model training. To ensure efficiency, we design $X_1$ and $X_2$ to be aligned at the same time lag, while noting that the framework naturally extends to cross-lagged interactions across different time steps.

**Efficient computation of temporal RUS in high dimensions.** Computation of PID was limited to discrete and small support (Bertschinger et al., 2014; Griffith & Koch, 2014), or continuous but

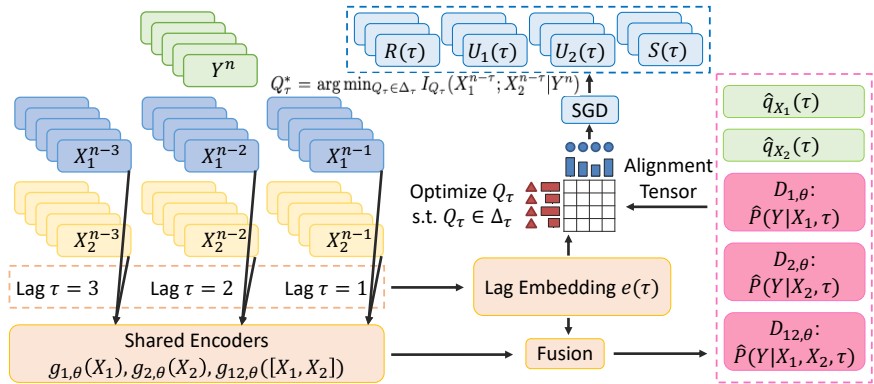

Figure 3: Schematic overview of the multi-scale BATCH estimator. The procedure consists of four stages: (1) encoding empirical datasets $\mathcal{D}_\tau = \left\{ (X_1^{n-\tau}, X_2^{n-\tau}, Y^n) \right\}_{\tau=1}^{3}$ with shared encoders $g$, with each lag $\tau$ further embedded by an encoder $e$ to produce $e(\tau)$, (2) training lag-conditioned discriminators $D_{1,\theta}, D_{2,\theta}, D_{12,\theta}$ to estimate $\hat{P}$, together with MLPs to generate embeddings for $\hat{q}$ at each $\tau$, (3) updating the alignment tensor to enforce marginal distribution matching between $Q_\tau$ and $\hat{P}$, yielding the optimized distribution $Q_\tau^*$, and (4) decomposing the resulting estimates of $Q_\tau^*$ and $\hat{P}$ into redundancy, uniqueness, and synergy sequence across all time lags.

low-dimensional variables (Pakman et al., 2021). The BATCH estimator (Liang et al., 2023) offers an effective solution for handling high-dimensional distributions: it parameterizes the distribution components of interest using neural networks and approximates the true distribution based on sub-sampled batches. The direct way to obtain temporal RUS values is to compute Eq. 5–7 for each $t \in [n-\tau, n]$, where all the distributional components of $P_\tau$ and $Q_\tau$ are estimated using the BATCH estimator. However, this can introduce significant computational overhead, as the optimization required to obtain $Q_\tau^*$ may need to be repeated at every step. To leverage the multitask nature of neural network backbones, we enhance the BATCH estimator by formalizing a multi-scale approach that trains a single model to predict the temporal RUS at multiple time lags, as shown in Figure 3.

For each lag $\tau$, we construct the empirical dataset $\mathcal{D}_\tau = \left\{ (X_1^{n-\tau}, X_2^{n-\tau}, Y^n) \right\}$, with $\tau$ chosen such that $n - \tau > 1$. We first estimate distributions $P_\tau(Y|X_1), P_\tau(Y|X_2), P_\tau(Y|X_1, X_2)$ by training lag-conditioned discriminators $D_1, D_2$, and $D_{12}$ for all $\tau$. For example, $D_{12}$ is defined as

$$\hat{P}(Y|X_1, X_2, \tau) = D_{12,\theta}\big(\phi(g_{12,\theta}([x_1; x_2]), e(\tau))\big), \tag{8}$$

where $g_{12,\theta}(\cdot)$ is the encoder, $\phi$ is the fusion operator, and $e(\tau)$ is learnable lag embeddings that encode temporal relationships. We then construct the optimal distribution $Q_\tau^*$ that satisfies $Q_\tau^* = \arg\min_{Q_\tau \in \Delta_\tau} I_{Q_\tau}(X_1^{n-\tau}; X_2^{n-\tau}|Y^n)$. This is achieved by constructing the alignment tensor $\text{align}_\tau \in \mathbb{R}^{N \times N \times C}$ that measures the compatibility between samples $i$ and $j$ from modalities $X_1$ and $X_2$, within class $k$, at lag $\tau$. Here, $N$ represents the batch size and $C$ the number of classes.

$$\text{align}_\tau[i, j, k] = \exp\left(\frac{\hat{q}_{X_1}^{(i,k,\tau)} \cdot \hat{q}_{X_2}^{(j,k,\tau)}}{\sqrt{d}}\right), \text{ where } \hat{q}_{X.}^{(i,k,\tau)} = \text{NN}_{\cdot}(\phi(g_{\cdot,\theta}(x_{\cdot,i}), e(\tau)))_k. \tag{9}$$

The alignment tensors are subsequently normalized via the Sinkhorn–Knopp algorithm (Knight, 2008) to enforce marginal-matching constraints, yielding the optimal distribution $Q_\tau^*$. We perform this step for all $\tau$ simultaneously by leveraging the efficient parallelism of tensor operations. Using Eq. 2–4, we can then compute temporal RUS values directly on high-dimensional embeddings.

## 3.2 MERGE: BUILDING RUS-AWARE MoE ROUTERS

We now discuss how to link routing decisions with information-theoretic principles to make the MoE interaction-aware. We begin by outlining the routing strategies for different interaction types. We then present the RUS-aware router, which incorporates the context of RUS sequences into MoE. Finally, we describe the auxiliary loss that enforces these routing strategies during training.

| RUS Values | Routing Strategies | Expert Types | Equivalent Fusion Types |
|---|---|---|---|
| High $R_{m_1,m_2}$ | Route tokens together | Regular Expert (shared) | Early Fusion |
| High $U_{m_1}$ | Diversify routing | Regular Expert | Late Fusion |
| High $S_{m_1,m_2}$ | Route tokens together | Cross-Modal Expert | Hybrid Fusion |

Table 1: Summary of our proposed routing strategies that leverage temporal RUS insights. Such a mechanism can also be viewed as a mixture of different fusion techniques (Baltrušaitis et al., 2018).

**Routing strategies.** Our model is based on the principle that different types of interactions call for different computational strategies: (1) When multiple modalities contain similar information (**R**), they can be routed to the same expert. This is supported by both theoretical insights from Kolmogorov complexity (Rissanen, 1978) and empirical findings in multimodal learning (Baltrušaitis et al., 2018; Ngiam et al., 2011). (2) When modalities contain distinct information (**U**), they should be routed to different experts to fully capture their uniqueness. Such routing diversification has also been highlighted in recent studies (Mustafa et al., 2022; Han et al., 2024) as a means of ensuring balanced expert utilization. (3) Since high synergy (**S**) benefits from explicit interaction modeling (Liang et al., 2023; Liu et al., 2018), we design dedicated synergy experts to capture cross-modal interactions. Each synergy expert consists of a cross-attention module followed by a feed-forward layer. In Table 1, we summarize the routing strategy and corresponding fusion techniques.

**RUS-aware router** is the central component that incorporates temporal RUS to guide MoE training. As illustrated in Figure 4, the attention mechanism focuses on pairwise redundancy and synergy between modalities, which weights the importance of different pairwise interactions based on the current token's characteristics. A GRU-based module captures the temporal dynamics of uniqueness, it further combines with attention-weighted interaction contexts. The final routing decision fuses token-specific representations with RUS-derived features. Formally,

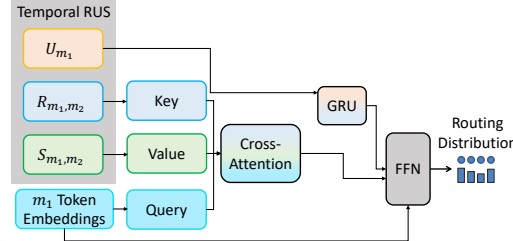

Figure 4: Router structure: routing decisions weighs RUS-derived features and token embeddings.

for each modality $m_1 \in \{1, 2, \ldots, M\}$ and all other modalities $m_2 \neq m_1$, let $\oplus$ denote concatenation. The RUS-aware router is formalized as

$$\text{RUSContext}_{m_1} = \text{Attention}\left(\text{Query}_{m_1}, \{[R_{m_1,m_2}, S_{m_1,m_2}]\}_{m_1 \neq m_2}\right) \oplus \text{GRU}(U_{m_1}), \quad (10)$$

$$\text{Logits}_{m_1} = \text{MLP}\left(\text{TokenFeatures}_{m_1} \oplus \text{RUSContext}_{m_1}\right). \quad (11)$$

This design ensures that routing decisions for modality $m_1$ are informed by its temporal interactions with all other modalities $m_2$ (captured by temporal RUS), not just its individual content.

**Auxiliary losses for multimodal interaction experts.** The training objective is formulated to ensure that the router adheres to the predefined principles. It combines the task-specific loss with a RUS-aware auxiliary loss, where the latter decomposes into components aligned with each interaction type. For redundancy, at each time step $t$, we employ Jensen–Shannon Divergence (JSD) to quantify the discrepancy between the routing distributions $P_{\text{router}}$ of $m_1$ and $m_2$. As shown in Eq. 12, minimizing $\mathcal{L}_{\text{redundancy}}$ encourages $m_1$ and $m_2$ to be routed to the same expert whenever their redundancy exceeds the threshold $\tau_R$.

$$\mathcal{L}_{\text{redundancy}} = \lambda_R \cdot \frac{1}{N_{\text{redundant}}} \sum_{(m_1,m_2,t):R_{m_1,m_2,t}>\tau_R} \text{JSD}(P_{\text{router}}^{(t,m_1)}, P_{\text{router}}^{(t,m_2)}). \quad (12)$$

Similarly, for uniqueness, as shown in Eq. 13, we encourage diversified routing of $m_1$ and $m_2$ whenever their uniqueness exceeds the threshold $\tau_U$ (uniqueness is modality-specific, not pairwise).

$$\mathcal{L}_{\text{uniqueness}} = -\lambda_U \cdot \frac{1}{N_{\text{unique}}} \sum_{(m_1,m_2,t):U_{m_1,t}>\tau_R \cap U_{m_2,t}>\tau_R} \text{JSD}(P_{\text{router}}^{(t,m_1)}, P_{\text{router}}^{(t,m_2)}). \quad (13)$$

We promote routing to synergy experts whenever synergistic information is high:

$$\mathcal{L}_{\text{synergy}} = \lambda_S \cdot \frac{1}{N_{\text{synergy}}} \sum_{(m_1,m_2,t):S_{m_1,m_2,t}>\tau_S} \left(1 - \frac{P_{\text{syn}}^{(t,m_1)} + P_{\text{syn}}^{(t,m_2)}}{2}\right). \quad (14)$$

Table 2: MERGE demonstrates superior results across different benchmarks and datasets. The best results are highlighted in **bold font**, and the second-best results are underlined.

| Datasets | PAMAP2 | | MIMIC-IV IHM | | MIMIC-IV LOS | | MOSI | | WESAD | | Opportunity |
|---|---|---|---|---|---|---|---|---|---|---|---|
| Metrics | Accuracy | F1 | AUROC | F1 | AUROC | F1 | Accuracy | AUROC | Accuracy | AUROC | Accuracy |
| Transformer | $82.48 \pm 1.26$ | $82.57 \pm 1.52$ | $80.18 \pm 0.17$ | $78.96 \pm 0.65$ | $75.46 \pm 0.46$ | $72.31 \pm 0.56$ | $68.39 \pm 0.36$ | $68.47 \pm 0.41$ | $53.84 \pm 1.27$ | $74.39 \pm 1.15$ | $81.59 \pm 0.74$ |
| mTAND | $74.62 \pm 0.53$ | $74.38 \pm 0.75$ | $80.89 \pm 0.33$ | $79.35 \pm 0.49$ | $77.34 \pm 0.29$ | $73.45 \pm 0.44$ | $70.07 \pm 0.47$ | $69.94 \pm 0.29$ | $48.22 \pm 0.88$ | $71.66 \pm 0.97$ | $70.26 \pm 2.03$ |
| MulT | $82.23 \pm 0.39$ | $81.87 \pm 0.43$ | $81.63 \pm 0.47$ | $81.55 \pm 0.53$ | $76.68 \pm 0.93$ | $72.52 \pm 0.77$ | $68.80 \pm 0.78$ | $69.05 \pm 0.83$ | $47.63 \pm 0.43$ | $71.43 \pm 0.33$ | $72.61 \pm 0.89$ |
| MISTS | $85.34 \pm 0.78$ | $85.79 \pm 0.31$ | $82.21 \pm 0.75$ | $80.56 \pm 0.72$ | $77.49 \pm 0.65$ | $73.86 \pm 0.35$ | $69.42 \pm 0.32$ | $69.32 \pm 0.51$ | $51.72 \pm 0.71$ | $73.29 \pm 0.61$ | $79.36 \pm 1.41$ |
| FuseMoE | $\underline{87.74} \pm 0.49$ | $\underline{86.51} \pm 1.17$ | $82.33 \pm 0.45$ | $81.64 \pm 0.58$ | $\underline{81.74} \pm 0.65$ | $\mathbf{75.18} \pm 0.41$ | $\mathbf{75.65} \pm 1.56$ | $\mathbf{78.43} \pm 0.97$ | $\underline{53.92} \pm 1.14$ | $\underline{76.31} \pm 0.99$ | $\underline{83.15} \pm 1.12$ |
| I2MoE | $84.55 \pm 0.64$ | $84.24 \pm 0.33$ | $\underline{83.28} \pm 0.27$ | $\underline{82.59} \pm 0.30$ | $79.88 \pm 1.08$ | $74.36 \pm 0.89$ | $71.91 \pm 2.20$ | $74.88 \pm 1.33$ | $52.64 \pm 1.06$ | $75.52 \pm 1.34$ | $82.16 \pm 2.57$ |
| MERGE | $\mathbf{91.37} \pm 1.38$ | $\mathbf{90.44} \pm 1.02$ | $\mathbf{85.40} \pm 0.24$ | $\mathbf{84.97} \pm 0.35$ | $\mathbf{81.88} \pm 0.18$ | $\underline{74.43} \pm 0.46$ | $\underline{72.04} \pm 1.99$ | $\underline{77.25} \pm 0.91$ | $\mathbf{55.74} \pm 1.99$ | $\mathbf{77.34} \pm 1.00$ | $\mathbf{84.32} \pm 1.33$ |

The overall objective combines the task-specific loss and the interaction-aware losses defined in Eq. 12–14. Importantly, the train–test split used for temporal RUS computation is aligned with that of MERGE's training, ensuring that the estimated interactions directly facilitate the learning of the corresponding tokens. Note that, the RUS estimation is performed separately from the MERGE training phase. While it is possible to build an end-to-end framework that jointly learns temporal RUS values and MoE model, we intentionally decouple these two components for the following reasons: (1) the multi-scale batch estimator is designed to reflect intrinsic information-theoretic structure of the dataset. It should not be optimized to minimize task-specific losses, doing so would entangle the RUS values with downstream objectives, undermining their role as task-agnostic signals of temporal multimodal interactions. (2) Since temporal RUS is a property of the dataset, it only needs to be computed *once*. The resulting values can then be cached and reused across any number of downstream tasks for this dataset. We expect MERGE to leverage these implicit interaction dynamics, thereby enhancing both the performance and the interpretability of downstream tasks.

## 4 EXPERIMENTS

**Overview.** We conduct comprehensive empirical evaluations of MERGE, aiming to answer the following questions: (1) What insights can temporal RUS provide in different scenarios? (2) Can leveraging temporal RUS improve MoE training accuracy? (3) How do temporal RUS influence task outcomes? (4) How can we interpret the learned routing assignments? Our evaluation spans diverse domains, including healthcare, activity recognition, and affective computing. In these evaluations, we incorporate a wide range of heterogeneous input streams, including sensors, wearables, medical imaging, text, and ECG, as distinct modalities to demonstrate MERGE's ability to capture complex temporal interactions among them and its effectiveness in massively multimodal settings.

### 4.1 EXPERIMENT SETUP

**Dataset information.** **PAMAP2** (Reiss & Stricker, 2012) is a physical activity monitoring benchmark that includes recordings of 18 activities, carried out by 9 participants equipped with 3 inertial measurement units and a heart rate monitor. The **MIMIC-IV** (Johnson et al., 2023) ecosystem contains deidentified clinical data from ICU patients, including lab measurements, vitals, notes, and chest X-rays. We consider two tasks: in-hospital mortality (IHM) and length-of-stay (LOS) prediction. **CMU-MOSI** (Zadeh et al., 2016) is a multimodal dataset of 2,199 annotated YouTube vlog opinion segments with audio, video, and text, used here for binary sentiment classification. **WESAD** (Schmidt et al., 2018) is a wearable sensor dataset from 15 subjects in a lab study, with physiological and motion data from chest- and wrist-worn devices. It supports affect recognition across six states, including stress, amusement, and meditation. **Opportunity** (Chavarriaga et al., 2013) provides multi-modal sensor recordings from wearable, object, and ambient devices, capturing natural and scripted daily activities of 4 subjects in a realistic home-like environment.

**Baselines.** To validate the effectiveness of MERGE in enhancing multimodal learning, we benchmark it against a diverse set of state-of-the-art models. The comparison covers: (1) standard monolithic backbones, including multi-head self-attention based Transformers (Vaswani et al., 2017) and multi-time attention for irregular time series (mTAND) (Shukla & Marlin, 2021); (2) task-specific fusion models, such as MulT (Tsai et al., 2019) and MISTS (Zhang et al., 2023); and (3) alterna-

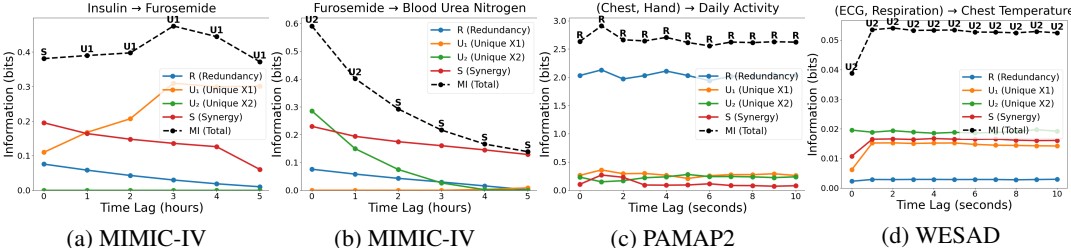

Figure 5: Insights from temporal RUS values across different applications include: (a) insulin and furosemide exhibit a strong synergistic effect at the time of administration, while insulin's unique effect becomes more pronounced later; (b) as time progresses after furosemide administration, its physiological impact increases; (c) in activity recognition, chest and hand motion display coupled movements during locomotion, reflecting strong redundancy; and (d) in physiological monitoring, ECG and respiration signals from one second prior provide better predictions of current chest temperature, capturing the natural response delay to stimuli.

tive multimodal MoE approaches, including FuseMoE (Han et al., 2024) that does not incorporate multimodal interaction, and I2MoE (Xin et al., 2025) that uses interaction-specific experts.

**Implementation.** We split PAMAP2, WESAD, and Opportunity into training, validation, and test sets by subject. For MIMIC-IV, we use the first 48 hours of each ICU stay as input and randomly assign the stays to training, validation, or test splits. For MOSI, we adopt the official train/val/test splits provided in Multibench (Liang et al., 2021). Unlike the other datasets, MIMIC-IV and MOSI contain many more sequences, and the target for each sequence is a single static label rather than a label sequence. When computing RUS values for these datasets, we focus on a single time step $n$, taking $Y^n$ as the label for each input sequence while still considering multiple lags $\tau$ from the end of the sequence. Because the RUS computation aggregates across multiple sequences, the resulting variety of labels ensures that the estimation of $P(Y^n)$ does not produce a degenerate distribution.

## 4.2 Performance and Insights of MERGE

**MERGE demonstrates superior performance.** In Table 2, we compare the performance of MERGE against various baseline models across 6 multimodal benchmark tasks. Results (averaged over 5 random runs) demonstrate that MERGE consistently achieves superior performance, attaining the best results in nearly all metrics. In particular, for healthcare (MIMIC-IV) and affective computing tasks (MOSI and WESAD), MERGE outperforms task-specific fusion methods such as MulT and MISTS by a significant margin. Against multimodal MoE baselines including FuseMoE and I2MoE, MERGE also achieves improved results. These findings highlight that leveraging interaction dynamics across modalities can effectively enhance model performance, and that MERGE generalizes well across diverse multimodal learning tasks.

**Qualitative examples of temporal RUS.** To understand the reason behind the improved performance, we examine the temporal RUS values obtained across different tasks to assess whether they provide rich and interpretable insights. For example, the changing relationship between insulin and furosemide over time in MIMIC-IV, shown in Fig.5 (a), illustrates the interplay between their immediate and delayed physiological effects. At the time of simultaneous administration, a synergistic effect emerges, likely driven by their concurrent but opposing influences on blood glucose and potassium levels. However, when insulin is administered 1 hour earlier or more, its uniqueness becomes much stronger, reflecting its relatively rapid onset of action and peak effect occurring within 2–3 hours. As shown in Fig.5 (b), at the time of Furosemide administration, its immediate diuretic effect is unique to the drug and is not yet fully captured in the Blood Urea Nitrogen (BUN) levels. Over time, however, the synergy between Furosemide and BUN increases, reflecting the delayed manifestation of the drug's physiological effects in blood chemistry.

For activity recognition on PAMAP2, during activities such as walking, running, or climbing stairs, the natural swinging of the arms is directly coupled with the torso and chest movements. This relationship remains consistent over time, leading to a high redundancy pattern, as shown in Fig. 5 (c). Lastly, examining the dynamics between ECG, respiration, and chest temperature in Fig. 5 (d)

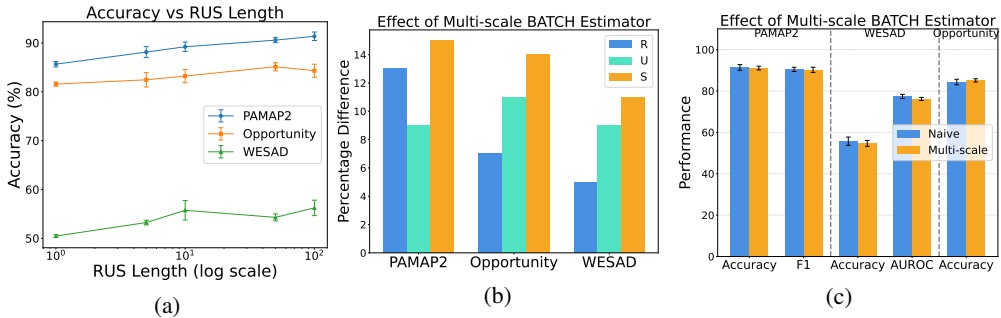

Figure 6: (a) Increasing temporal RUS length incorporated into MoE training improves performance; (b) discrepancy between the multi-scale BATCH estimator versus step-wise RUS computation; (c) performance differences across tasks after applying the multi-scale BATCH estimator.

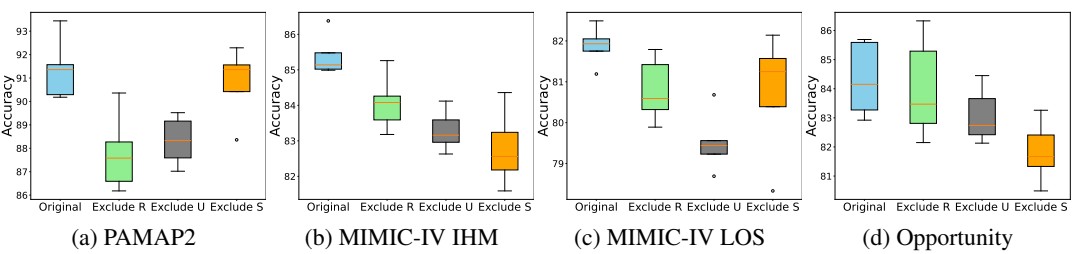

Figure 7: Performance change after removing each auxiliary loss term from Eq.12 to 14. The results show that all interactions contribute effectively to enhancing performance across nearly all settings.

reveals an initial increase in synergy and respiration uniqueness on WESAD dataset. This arises because the one-second delay aligns with the time it takes for the body's stress response to begin manifesting as a change in skin temperature. After this brief adjustment period, the relationships stabilize, reflecting the steady state of the physiological response. These examples clearly demonstrate that temporal RUS can capture unique interaction dynamics for improved model performance.

### 4.3 IN-DEPTH ANALYSIS OF MERGE

**Impact of temporal RUS length.** We investigate how the length of temporal RUS incorporated into MoE training affects performance. Our intuition is that a longer temporal range captures richer interaction information, thereby improving results. We first increase the maximum time lag of the temporal RUS from 1 to 10. We do not extend it further, as estimating too many steps simultaneously can degrade the performance of the multi-scale BATCH estimator. To further extend sequence length, we repeat temporal RUS segments to form longer sequences. As shown in Fig. 6 (a), both strategies lead to performance gains. Increasing the maximum time lag provides a broader view of the temporal trajectory. Meanwhile, segment repetition appears to strengthen the model's ability to recognize important modality interactions, improve context modeling, and reduce gradient variance. A similar observation that longer sequences with consistent patterns improve generalization has also been discussed in prior works (Mustafa et al., 2022; Lai et al., 2018).

**Efficiency of multi-scale BATCH estimator.** The proposed multi-scale BATCH estimator achieves $\tau$-fold speedup in temporal RUS estimation while maintaining parameter efficiency. We compare the estimated RUS trajectories against the naive step-wise approach. As shown in Fig.6 (b), we report the percentage difference across tasks, averaged over all time lags. Although the magnitudes of individual interactions differ slightly, the discrepancies remain relatively small. Moreover, these differences do not lead to notable performance fluctuations, as demonstrated in Fig.6 (c).

**Impact of each auxiliary loss.** We further investigate the contribution of each auxiliary loss term to performance across tasks. To this end, we perform an ablation study by removing each auxiliary loss term from Eq.12 to 14, thereby disabling the corresponding routing mechanism. As shown in Fig.7,

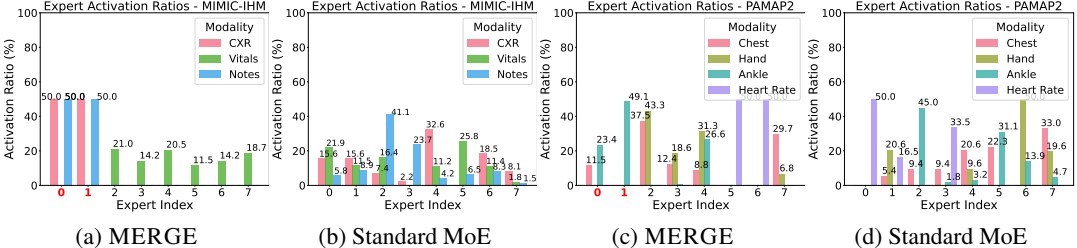

Figure 8: Comparison of routing distributions between MERGE and standard MoE. The activation ratio is defined by the percentage of tokens from a specific modality that are routed to each expert. In (a) and (c), expert indices highlighted in red denote synergy experts. The plots show that MERGE adheres to the proposed routing principles and provides enhanced interpretability.

uniqueness emerges as a critical factor for many tasks, while MIMIC-IV IHM and Opportunity are primarily synergy-driven, and PAMAP2 shows greater dependence on redundancy.

**Expert routing analysis.** Finally, we explicitly examine how MERGE influences multimodal token routing distributions. We compare MERGE with a standard MoE using top-$k$ routing, as shown in Fig.8. The distributions are generated from models that achieve the best performance for each task. In (a), RUS values show there is strong synergy between CXR and clinical notes, while vitals remain relatively independent. The router captures this by directing corresponding tokens to specialized experts, enabling them to be trained more effectively together. Likewise, in (c), chest and hand signals (with strong redundancy) are routed to the same expert, whereas modality pairs with high synergy or uniqueness are also handled according to their interaction type. In contrast, the standard MoE exhibits no such structured routing: experts process arbitrary combinations of modalities, resulting in unorganized assignments that hinder both efficient learning and interpretability.

## 5   CONCLUSION AND FUTURE WORK

We introduced MERGE, a novel MoE architecture that integrates temporal multimodal interactions into model training. By decomposing multi-source directed information into temporal redundancy, uniqueness, and synergy, our approach captures rich evolving dynamics that effectively guide MoE routing in massively multimodal applications. We further developed scalable estimators suitable for multi-time-lag settings. MERGE achieves substantial performance gains while preserving expert-level interpretability, making it a compelling design choice for future multimodal foundation models. Looking ahead, promising directions include extending the framework to capture more general spatio-temporal dynamics and investigating its applicability to multimodal and multitask learning scenarios. We would also like to extend the design philosophy of leveraging temporal multimodal interaction to guide expert routing to LLM-MoE frameworks. For instance, incorporating multimodal inputs with known temporal RUS values could potentially improve fine-tuning of MoE-based LLMs by helping the model identify the appropriate experts. Furthermore, we believe the current MERGE design shows even greater potential when applied to large-scale VLMs or world-models, where temporal multimodal interaction dynamics remain highly informative and can meaningfully improve real-world performance.

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

## A  ADDITIONAL RELATED WORKS

Partial Information Decomposition (PID) (Williams & Beer, 2010; Bertschinger et al., 2014) has emerged as a formal way to quantify multimodal interactions by measuring how the total information between two modalities $(x_1, x_2)$ useful for a task $y$ can be decomposed into redundant $(R)$, unique $(U)$, and synergistic $(S)$ parts. Redundancy measures the common information between two modalities, uniqueness measures the useful information in a modality not present in the others, and synergy measures the new information that arises only when both modalities are fused. Specifically, given unimodal marginal distributions $p(x_1, y)$ and $p(x_2, y)$ over each modality and the multimodal joint distribution $p(x_1, x_2, y)$, a formal definition is

$$R = \max_{q \in \Delta_p} I_q(X_1; X_2; Y), \tag{15}$$

$$U_1 = \min_{q \in \Delta_p} I_q(X_1; Y | X_2), \quad U_2 = \min_{q \in \Delta_p} I_q(X_2; Y | X_1), \tag{16}$$

$$S = I_p(X_1, X_2; Y) - \min_{q \in \Delta_p} I_q(X_1, X_2; Y), \tag{17}$$

where $\Delta_p := \{q \in \Delta : q(x_i, y) = p(x_i, y), \forall y \in \mathcal{Y}, x_i \in \mathcal{X}_i, i \in [2]\}$ characterizes the set of *marginal-matching* joint distributions, and $I_q$ is the mutual information (MI) over these joint distribution $q(x_1, x_2, y)$. The key lies in optimizing $q \in \Delta_p$ such that the marginals $q(x_1, y) = p(x_1, y)$ and $q(x_2, y) = p(x_2, y)$ are preserved, while relaxing the coupling between $x_1$ and $x_2$; that is, $q(x_1, x_2)$ is not necessarily equal to $p(x_1, x_2)$. The intuition behind this is that redundancy and uniqueness should be identifiable given access only to $p(x_1, y)$ and $p(x_2, y)$, and thus should depend solely on $q \in \Delta_p$. In contrast, synergy inherently depends on the joint distribution $p(x_1, x_2)$, which is reflected in Eq.17 relying on the full distribution $p$.

**Multimodal Fusion.**    Early approaches to multimodal fusion employed kernel-based methods (Bucak et al., 2013; Chen et al., 2014; Poria et al., 2015), graphical models (Nefian et al., 2002; Garg et al., 2003; Reiter et al., 2007), and neural networks (Ngiam et al., 2011; Gao et al., 2015; Nojavanasghari et al., 2016). With advances in deep learning, more sophisticated fusion strategies have emerged. Tensor-based methods (Zadeh et al., 2017; Liu et al., 2018) perform outer-product fusion to capture multiplicative interactions, while attention-based approaches (Rahman et al., 2020; Yang & Wu, 2021) generate cross-modal displacement vectors through self-attention mechanisms. The Multimodal Transformer (MulT) (Tsai et al., 2019) introduced cross-modal attention blocks for word-level alignment across vision, language, and audio. In clinical settings, late fusion approaches (Khadanga et al., 2019; Deznabi et al., 2021) concatenate embeddings from pre-trained encoders, while Soenksen et al. (2022) developed a generalizable pipeline for electronic health records spanning four modalities through direct concatenation followed by gradient boosting. More recently, MISTS (Zhang et al., 2023) extended cross-modal attention with multi-time attention modules to handle temporal irregularities.

Table 3: Comparison between Multimodal MoE and MERGE for massively multimodal learning.

| Aspect | Multimodal MoE | MERGE |
|---|---|---|
| Routing Basis | Expert-token similarity | Temporal RUS + Similarity |
| Modality Scalability | $O(M^2)$ implicit | Sub-linear (RUS-guided) |
| Cross-Modal Awareness | Implicit | Explicit pairwise temporal interaction |
| Expert Specialization | Emergent from training | Principled (R/U/S-specific) |
| Temporal Dynamics | Single time point | Multi-lag temporal interaction |
| Routing Decision | Black-box | Interpretable RUS values |
| Computational Efficiency | Standard sparsity | RUS-optimized sparsity |
| Fusion Strategy | Fixed | Adaptive mixture of fusion types |

**MoE-Based Multimodal Fusion.**    MoE architectures have emerged as a promising paradigm for multimodal fusion due to their ability to scale efficiently while enabling expert specialization. LIMoE (Mustafa et al., 2022) pioneered large-scale multimodal MoE by processing both images and text through a shared sparse transformer with contrastive learning, demonstrating that experts naturally specialize in different modalities through entropy-based regularization. FuseMoE (Han

et al., 2024) addressed the "FlexiModal" setting with irregularly sampled and missing modalities, introducing a Laplace gating function with theoretical convergence guarantees superior to softmax routing. Building on this, Flex-MoE (Yun et al., 2024) proposed a missing modality bank and dual-router design ($\mathcal{G}$-Router for generalized knowledge, $\mathcal{S}$-Router for modality-specific combinations) to handle arbitrary modality availability. Hierarchical MoE (Nguyen et al., 2024) demonstrated that Laplace gating at two hierarchical levels eliminates undesirable parameter interactions, accelerating expert convergence in multimodal tasks. $I^2$MoE (Xin et al., 2025) incorporated interaction-type awareness by assigning separate experts to redundancy, uniqueness, and synergy interactions using weakly supervised losses derived from Partial Information Decomposition. ConfSMoE (Zheng et al., 2025) addressed expert collapse through confidence-guided gating that detaches routing scores from task confidence, providing theoretical insights into why experts fail to specialize under standard softmax routing. Despite these advances, existing multimodal MoE approaches face limitations in *massively multimodal* settings involving dozens to hundreds of heterogeneous input streams. We summarize these challenges and compare key aspects of multimodal MoE with MERGE in Table 3.

## B  DATASET INFORMATION AND PROCESSING DETAILS

### B.1  MIMIC-IV

The Medical Information Mart for Intensive Care (MIMIC-IV) ecosystem comprises deidentified clinical databases of patients admitted to the emergency department or intensive care unit (ICU) at Beth Israel Deaconess Medical Center. We use lab measurements, vital signs, and radiology notes from MIMIC-IV and MIMIC-IV-Notes (Johnson et al., 2023), along with chest X-rays from MIMIC-CXR (Johnson et al., 2019a;b), to construct temporal sequences of ICU stays. Input signals are categorized into three modalities: labs/vitals, clinical notes, and chest X-rays. We extract embeddings from clinical notes using BioBERT (Lee et al., 2020) and from chest X-rays using a DenseNet (Huang et al., 2017) pre-trained on CheXpert (Irvin et al., 2019). We consider two prediction tasks: in-hospital mortality (IHM) and length-of-stay (LOS). For IHM, a stay is labeled as mortality if the patient died after the first 48 hours, otherwise as survival. For LOS, the goal is to predict whether the patient will leave the ICU alive in less than 96 hours. The train, validation, and test splits consist of 32,435, 6,950, and 6,952 stays, respectively.

### B.2  CMU-MOSI

CMU-MOSI (Zadeh et al., 2016) is a multimodal dataset consisting of 2,199 opinion video segments, including speech, visual gestures, and audio, drawn from 93 YouTube vlog videos and annotated for sentiment intensity. We use the preprocessed version from MultiBench (Liang et al., 2021) and focus on the binary sentiment classification task (positive vs. negative). The modalities considered are vision, text, and audio. The train, validation, and test splits contain 1,283, 214, and 686 segments, respectively.

### B.3  WESAD

WESAD (Schmidt et al., 2018) is a multimodal dataset for wearable stress and affect detection, collected from 15 participants in a controlled laboratory study. It includes physiological and motion data from chest- and wrist-worn devices, covering signals such as electrodermal activity (EDA), electrocardiogram (ECG), acceleration, temperature, blood volume pulse (BVP), and respiration. The task is affect recognition across six states: transient, baseline, stress, amusement, meditation, and unknown. We group the signals into two modalities: chest signals and wrist signals. The dataset is split by subject, with 10, 2, and 3 subjects allocated to the train, validation, and test sets, respectively.

### B.4  OPPORTUNITY

The Opportunity dataset (Chavarriaga et al., 2013) comprises a comprehensive human activity recognition corpus collected in a sensor-rich kitchen environment, featuring four subjects performing activities of daily living (ADL). The dataset contains 20 experimental sessions, with each subject completing 5 ADL runs (S1-ADL1 through S4-ADL5), while drill sessions were excluded from our

experimental protocol. Data collection was conducted at a 30 Hz sampling frequency, yielding 250 feature columns encompassing 243 sensor measurements and 7 activity labels. The activity recognition task focuses on 5 high-level activity classes: Relaxing (101), Coffee time (102), Early morning (103), Cleanup (104), and Sandwich time (105). The preprocessing pipeline implemented a subject-based train/validation/test split protocol to prevent temporal data leakage, allocating subjects 1-3 (runs ADL1-ADL4) for training (342,535 samples), subjects 1-3 (run ADL5) for validation (76,399 samples), and subject 4 (all runs) for testing (111,644 samples). The 243 sensor features were systematically categorized into seven distinct modalities based on anatomical location and sensor type: Torso (19 sensors), Arms (58 sensors), Legs (52 sensors), Shoes (32 sensors), Objects (35 sensors), Environment (34 sensors), and Location (12 sensors).

### B.5 PAMAP2

The PAMAP2 Physical Activity Monitoring dataset (Reiss & Stricker, 2012) comprises a comprehensive multimodal sensor corpus collected from nine participants performing 18 distinct physical activities in naturalistic settings. The dataset incorporates heterogeneous sensor modalities including three Colibri wireless Inertial Measurement Units (IMUs) positioned at anatomically strategic locations: dominant wrist, chest, and dominant ankle, sampling at 100 Hz, complemented by a heart rate monitor operating at approximately 9 Hz. Each experimental session captures 54-dimensional feature vectors encompassing temporal timestamps, activity labels, and 52 attributes of raw sensory measurements spanning accelerometry (both 16g and 6g ranges), gyroscopy, magnetometry, and quaternion orientation data. The activity recognition framework targets 18 physical activities including fundamental locomotion patterns (lying, sitting, standing, walking, running), complex motor tasks (cycling, stair navigation), daily living activities (computer work, household tasks), and recreational pursuits (soccer, rope jumping). Similar to the Opportunity dataset, we implemented a subject-based partitioning strategy to ensure temporal independence and prevent data leakage, allocating subjects 1-6 for training, subject 7 for validation, and subjects 8-9 for testing. The 52-dimensional sensor feature space was systematically organized into four modalities: chest, hand, ankle, and heart rate.

## C   DETAILED PROCEDURE OF MULTI-SCALE BATCH ESTIMATOR

---

**Algorithm 1** Phase 1: Multi-Task Discriminator Training

---

**Require:** Multi-lag datasets $\{\mathcal{D}_\tau\}_{\tau=0}^{K-1}$, lag weights $\{w_\tau\}$
 1: Initialize shared backbone networks and lag-specific components
 2: **for** epoch = 1 to $N_{\text{disc}}$ **do**
 3:     **for** step = 1 to steps_per_epoch **do**
 4:         Sample lag $\tau \sim \text{Categorical}(\{w_\tau\})$
 5:         Sample batch $(x_{1,\tau}^{(b)}, x_{2,\tau}^{(b)}, y^{(b)}) \sim \mathcal{D}_\tau$
 6:         **Forward pass with lag conditioning:**
 7:         $\text{logits}_1 \leftarrow D_{1,\theta}(x_{1,\tau}^{(b)}, \tau)$
 8:         $\text{logits}_2 \leftarrow D_{2,\theta}(x_{2,\tau}^{(b)}, \tau)$
 9:         $\text{logits}_{12} \leftarrow D_{12,\theta}([x_{1,\tau}^{(b)}; x_{2,\tau}^{(b)}], \tau)$
10:         **Compute discriminator loss:**
11:         $\mathcal{L}_{\text{batch}} \leftarrow H(y^{(b)}, \text{logits}_1) + H(y^{(b)}, \text{logits}_2) + H(y^{(b)}, \text{logits}_{12})$
12:         Update $\theta$ via $\nabla_\theta \mathcal{L}_{\text{batch}}$
13:     **end for**
14: **end for**
15: **return** Trained discriminator parameters $\theta_{\text{disc}}$

---

---

**Algorithm 2** Phase 2: Multi-Task Alignment for $Q$

---

**Require:** Multi-lag datasets $\{\mathcal{D}_\tau\}_{\tau=0}^{K-1}$, lag weights $\{w_\tau\}$, trained discriminators $\theta_{\text{disc}}$
 1: Initialize lag-conditioned Alignment module with parameters $\theta_{\text{align}}$
 2: Freeze discriminator parameters $\theta_{\text{disc}}$
 3: **for** epoch = 1 to $N_{\text{align}}$ **do**
 4:    **for** step = 1 to steps_per_epoch **do**
 5:       Sample lag $\tau \sim \text{Categorical}(\{w_\tau\})$
 6:       Sample batch $(x_{1,\tau}^{(b)}, x_{2,\tau}^{(b)}, y^{(b)}) \sim \mathcal{D}_\tau$
 7:       **Compute lag-conditioned embeddings:**
 8:       $q_{X_1} \leftarrow \phi_1(g_{1,\theta}(x_{1,\tau}^{(b)}), e(\tau))$
 9:       $q_{X_2} \leftarrow \phi_2(g_{2,\theta}(x_{2,\tau}^{(b)}), e(\tau))$
10:       **Compute lag-specific alignment:**
11:       $\text{align}_\tau[i,j,k] \leftarrow \exp\left(\frac{q_{X_1}^{(i,k)} \cdot q_{X_2}^{(j,k)}}{\sqrt{d}}\right)$
12:       **Sinkhorn normalization with lag-specific discriminators:**
13:       $P_{Y|X_1,\tau} \leftarrow \text{Softmax}(D_{1,\theta_{\text{disc}}}(x_1, \tau))$
14:       $P_{Y|X_2,\tau} \leftarrow \text{Softmax}(D_{2,\theta_{\text{disc}}}(x_2, \tau))$
15:       **for** $k = 1$ to $C$ **do**
16:          **for** iter = 1 to max_iter **do**
17:             $Q_\tau \leftarrow$ Apply Sinkhorn updates on $\text{align}_\tau[:,:,k]$ using $P_{Y|X_1,\tau}$ and $P_{Y|X_2,\tau}$
18:             Check convergence
19:          **end for**
20:       **end for**
21:       **Compute alignment loss:**
22:       $\mathcal{L}_{\text{align}} \leftarrow \text{AlignmentLoss}(Q_\tau, x_{1,\tau}^{(b)}, x_{2,\tau}^{(b)}, y^{(b)})$
23:       Update alignment parameters $\theta_{\text{align}}$ via $\nabla_{\theta_{\text{align}}}\mathcal{L}_{\text{align}}$
24:    **end for**
25: **end for**
26: **return** Trained alignment parameters $\theta_{\text{align}}$

---

---

**Algorithm 3** Phase 3: Multi-Lag RUS Computation

---

**Require:** Multi-lag datasets $\{\mathcal{D}_\tau\}_{\tau=0}^{K-1}$, trained discriminators $\theta_{\text{disc}}$, trained alignment module $\theta_{\text{align}}$
 1: **for** $\tau \in \{0, 1, \ldots, K-1\}$ **do**
 2:    **Compute mutual information (MI) terms using trained discriminators:**
 3:    $\text{MI}_\tau(Y; X_1) \leftarrow \text{EstimateMI}(D_{1,\theta_{\text{disc}}}(\cdot, \tau), \mathcal{D}_\tau)$
 4:    $\text{MI}_\tau(Y; X_2) \leftarrow \text{EstimateMI}(D_{2,\theta_{\text{disc}}}(\cdot, \tau), \mathcal{D}_\tau)$
 5:    $\text{MI}_\tau(Y; X_1, X_2) \leftarrow \text{EstimateMI}(D_{12,\theta_{\text{disc}}}(\cdot, \tau), \mathcal{D}_\tau)$
 6:    **Compute optimal alignment distribution:**
 7:    $Q_\tau^* \leftarrow \text{ComputeOptimalAlignment}(\theta_{\text{align}}, \mathcal{D}_\tau, \tau)$
 8:    $\text{MI}_{Q_\tau^*}(Y; X_1, X_2) \leftarrow \text{EstimateMI}(Q_\tau^*, \mathcal{D}_\tau)$
 9:    **Decompose into RUS components:**
10:    $R_\tau \leftarrow \text{MI}_\tau(Y; X_1) + \text{MI}_\tau(Y; X_2) - \text{MI}_{Q_\tau^*}(Y; X_1, X_2)$ {Redundancy}
11:    $U_{1,\tau} \leftarrow \text{MI}_{Q_\tau^*}(Y; X_1, X_2) - \text{MI}_\tau(Y; X_2)$ {Uniqueness $X_1$}
12:    $U_{2,\tau} \leftarrow \text{MI}_{Q_\tau^*}(Y; X_1, X_2) - \text{MI}_\tau(Y; X_1)$ {Uniqueness $X_2$}
13:    $S_\tau \leftarrow \text{MI}_\tau(Y; X_1, X_2) - \text{MI}_{Q_\tau^*}(Y; X_1, X_2)$ {Synergy}
14: **end for**
15: **return** Temporal RUS components $\{R_\tau, U_{1,\tau}, U_{2,\tau}, S_\tau\}_{\tau=0}^{K-1}$

---

# D  MODEL ARCHITECTURE DETAILS

## D.1  MODALITY-SPECIFIC ENCODER

| Component | Configuration |
|---|---|
| Input Projection Layer | Linear transformation from modality dimension $D_m$ to model dimension $d_{model}$ |
| Feature Scaling | Input embeddings scaled by $\sqrt{d_{model}}$ for training stability |
| Temporal Convolution (Optional) | Two 1D convolutional layers with residual connections for temporal feature extraction |
| Convolutional Architecture | 1D convolution: $d_{model} \rightarrow d_{model}$ with kernel size $k$ |
| Batch Normalization | Applied after each convolutional layer |
| Activation Function | ReLU activation for non-linearity |
| Self-Attention Layers | Multi-layer transformer encoder |
| Output Normalization | Layer normalization applied to final embeddings |

Table 4: Modality-Specific Encoder Architecture Components

| Parameter | Value | Description |
|---|---|---|
| Encoder Layers ($L_{enc}$) | 2 | Number of transformer encoder layers |
| Attention Heads ($H$) | 4 | Multi-head attention heads per layer |
| Feed-Forward Dimension ($d_{ff}$) | 256 | Inner dimension of position-wise FFN |
| Dropout Rate ($p_{drop}$) | 0.1 | Dropout probability for regularization |
| Convolution Kernel ($k$) | 3 | Temporal convolution kernel size |

Table 5: Modality-Specific Encoder Hyperparameters

## D.2  EXPERT NETWORKS

| Layer | Transformation |
|---|---|
| Input Layer | Linear transformation: $\mathbb{R}^{d_{model}} \rightarrow \mathbb{R}^{d_{expert}}$ |
| Activation | ReLU non-linear activation function |
| Regularization | Dropout with probability $p_{drop}$ |
| Output Layer | Linear transformation: $\mathbb{R}^{d_{expert}} \rightarrow \mathbb{R}^{d_{model}}$ |

Table 6: Feed-Forward Expert Network Architecture

| Component | Description |
|---|---|
| Multi-Head Self-Attention | Scaled dot-product attention with $H$ parallel heads |
| Attention Residual | Residual connection around attention sub-layer |
| Post-Attention Norm | Layer normalization after attention residual |
| Position-wise FFN | Two-layer feed-forward network with ReLU |
| FFN Residual | Residual connection around feed-forward sub-layer |
| Final Normalization | Layer normalization after FFN residual |
| Dropout Regularization | Applied to attention output and FFN output |

Table 7: Synergy Expert Network Architecture

| Parameter | Default | Description |
|---|---|---|
| Expert Hidden Dimension ($d_{expert}$) | 128 | Internal processing dimension |
| Synergy Expert Heads ($H_{syn}$) | 4 | Attention heads for synergy experts |
| Expert Dropout ($p_{expert}$) | 0.1 | Dropout rate within expert networks |
| Synergy Experts ($N_{syn}$) | 2 | Number of attention-based synergy experts |
| Total Experts ($N_{expert}$) | 8 | Total number of expert networks |

Table 8: Expert Network Hyperparameters

## D.3 RUS-AWARE ROUTER

| Component | Description |
|---|---|
| Token Processing | Linear projection: $\mathbb{R}^{d_{model}} \rightarrow \mathbb{R}^{d_{token}}$ with ReLU activation |
| Query Generation | Projects processed tokens to attention query space: $\mathbb{R}^{d_{token}} \rightarrow \mathbb{R}^{d_k}$ |
| RUS Key Projection | Maps pairwise RUS values to attention keys: $\mathbb{R}^2 \rightarrow \mathbb{R}^{d_k}$ |
| RUS Value Projection | Maps pairwise RUS values to attention values: $\mathbb{R}^2 \rightarrow \mathbb{R}^{d_v}$ |
| Temporal RUS Encoder | Single-layer GRU processes concatenated uniqueness and attention context |
| Routing MLP | Two-layer network outputs expert routing logits |

Table 9: RUS-Aware Routing Network Architecture

| Parameter | Value | Description |
|---|---|---|
| GRU Hidden Dimension ($d_{gru}$) | 64 | Temporal context encoding dimension |
| Token Processing Dimension ($d_{token}$) | 64 | Intermediate token representation size |
| Attention Key Dimension ($d_k$) | 32 | Query-key attention space dimension |
| Attention Value Dimension ($d_v$) | 32 | RUS value projection dimension |

Table 10: Routing Network Hyperparameters

# E   CHOICE OF HYPER-PARAMETERS IN AUXILIARY LOSSES

In this section, we discuss the roles of the hyperparameters $\tau$ and $\lambda$ that appear in the auxiliary loss terms in Eq.12–Eq.14, and provide guidance on how to tune them for improved performance in multimodal applications. The parameters $(\tau_R, \tau_U, \tau_S)$ serve as threshold values that determine when to activate different routing behaviors based on the RUS scores. Conceptually, they control the selectivity of the routing mechanism. If a threshold is set too low, almost all interactions will trigger specialized routing; if set too high, the routing reverts to a standard MoE regime and effectively ignores RUS-guided specialization. Empirically, synergistic interactions tend to be much rarer than redundancy or uniqueness in real-world multimodal data, so we typically set $\tau_S$ lower to avoid overly suppressing synergy-driven routing and to maintain a balanced distribution of routing patterns.

The parameters $(\lambda_R, \lambda_U, \lambda_S)$ regulate the strength of the auxiliary loss terms corresponding to each interaction type. These can be selected based on domain knowledge, e.g., whether a dataset is dominated by synergistic patterns, highly independent modalities, etc., or adjusted to explicitly encourage certain forms of interaction. They can also be made adaptive during training, with validation performance guiding the adjustment.

Although $\tau$ and $\lambda$ encode different aspects of the system, they are related in effect: increasing a threshold $\tau$ reduces how often a particular interaction pattern is activated, which is analogous to decreasing the corresponding $\lambda$, thereby weakening its influence in training.

In practice, we find that because the empirical distribution of RUS values is highly non-uniform, the thresholds $\tau$ (typically set as specific percentiles of their respective RUS distributions) are relatively insensitive to fine-grained tuning. Therefore, we precompute and fix $\tau$ based on dataset statistics and only tune the $\lambda$ values during model training.

# F    ADDITIONAL EXPERIMENTS

## F.1    SYNTHETIC TEMPORAL RUS

**Synthetic data generation**  To validate our multi-scale BATCH estimator framework for computing temporal RUS, we created a synthetic benchmark with known R/U/S values. This synthetic dataset consists of three time series: two independent source variables $X_1$ and $X_2$, and a target variable $Y$. Both $X_1$ and $X_2$ were independently sampled from $N(0,1)$. $Y$ was constructed with explicit temporal causal influences from both sources, where $X_1$ influences $Y$ with a time lag of 1 ($\tau = 1$) and $X_2$ influences $Y$ with a time lag of 2 ($\tau = 2$). At each time point $t$, $Y(t)$ was computed as: $Y(t) = \alpha \cdot X_1(t-1) + \alpha \cdot X_2(t-2) + \eta \cdot \epsilon(t)$, where $\epsilon(t)$ is noise sampled from $N(0,1)$. We followed the definition of direction information (Weissman et al., 2012) and information decomposition (Bertschinger et al., 2014) to compute temporal RUS values. Figure 9 demonstrates an example of the temporal RUS values we obtained for the synthetic experiments.

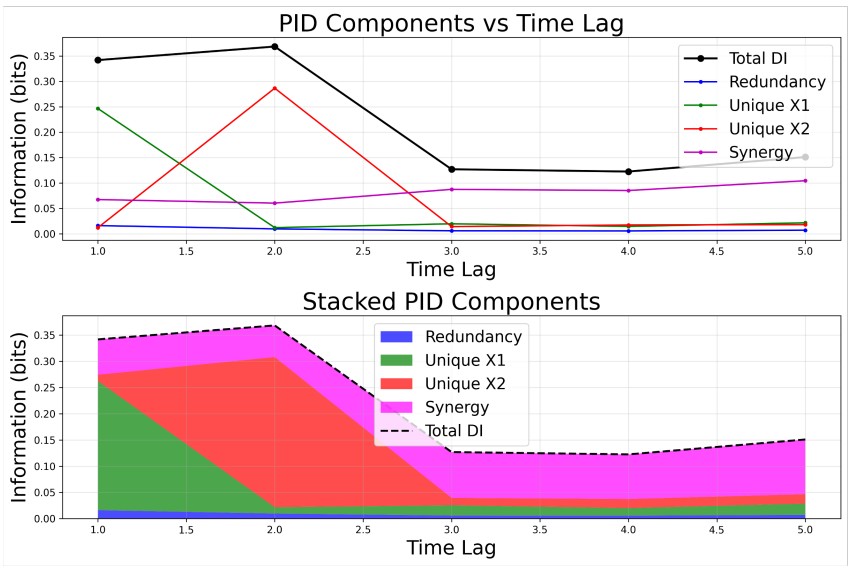

Figure 9: Ground truth temporal RUS of synthetic dataset: at $t = 1$, uniqueness of $X_1$ dominates the total directed information; at $t = 2$, uniqueness of $X_2$ dominates; from $t = 3$ and afterwards, synergy between $X_1$ and $X_2$ dominates the total directed information.

We compare the proposed multi-scale BATCH estimator with the step-wise temporal RUS computation using data with known RUS. Our evaluation spans multiple aspects, including varying sample sizes, different feature dimensions for $X_1/X_2$, encoder capacity within the BATCH estimator, and the number of Sinkhorn iterations used during optimization. Figure 10 to Figure 13 demonstrate these results: all experiments are averaged across 5 random runs with different time lag configurations; the reported R/U/S values are also aggregated across time lags.

## F.2    EFFECT OF BAD R/U/S ESTIMATIONS

We conducted additional experiments to assess how sensitive MERGE is to inaccuracies in the temporal RUS estimates, particularly in scenarios where the RUS signals are weak or noisy. To this end, we evaluated two variants of the model: (1) Noisy RUS: we injected Gaussian noise into the computed temporal RUS sequences to simulate unstable estimation. (2) Sparse / weakened RUS: we

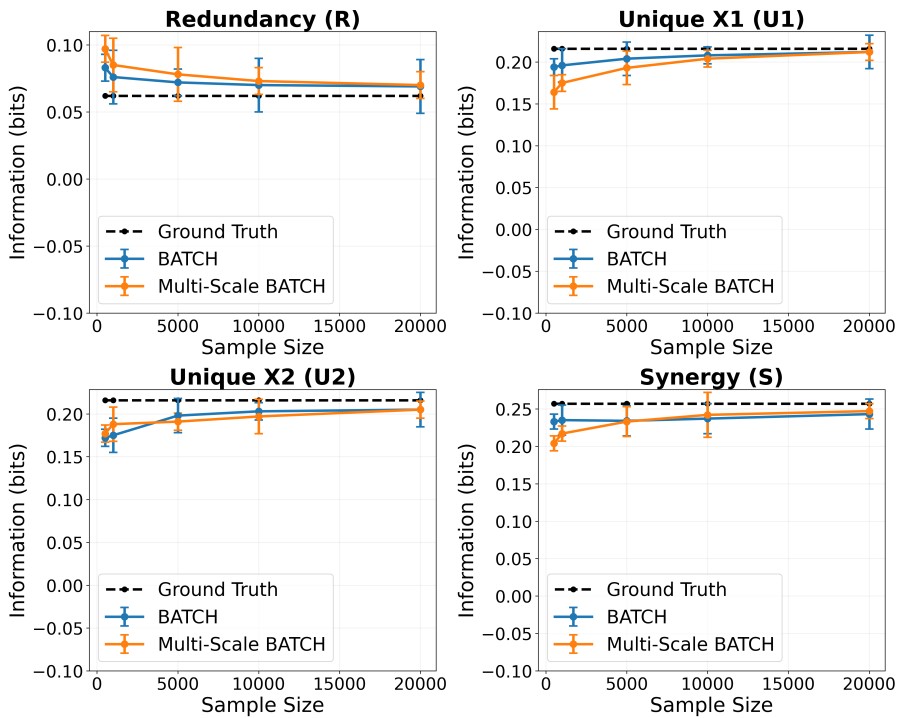

Figure 10: Ablation study of the multi-scale BATCH estimator under different sample sizes: we tested both methods on [500, 1000, 5000, 10000, 20000] samples. Results show that the multi-scale BATCH estimator leads to similar results with step-wise BATCH estimator. As sample size increases, both two methods achieve better approximations to ground truth R/U/S values.

zeroed out a majority of the temporal RUS sequence values, degrading them into sparse interaction patterns. Figure 14 demonstrates the results.

### F.3 TRAINING TIME & MAX GPU MEMORY

We summarize the total training cost, including both temporal RUS estimation and MERGE training, in terms of wall-clock training time and maximum GPU memory consumption across all datasets evaluated in the paper. Figure 15 presents scatter plots showing the distribution of training time versus peak GPU memory for each dataset.

### F.4 EFFECT OF CLASS IMBALANCE IN RUS ESTIMATION

For datasets with sequence-level labels such as MIMIC-IV and MOSI, the prediction task is performed at the full-sequence level. For example, in MIMIC-IV IHM, the model uses the first 48 hours of ICU measurements to predict in-hospital mortality. Accordingly, we compute temporal RUS at the final time step (i.e., sweeping lags backward from the end of the 48-hour window), since these values are the relevant values aligned with the prediction target. However, in situations such as class imbalance and label noise can influence both temporal RUS estimation and downstream routing behavior. As discussed in Section 4, each patient contains multiple measurements $\mathcal{D}_i \in \mathbb{R}^{d \times T}$, but is associated with only a single label $Y_i \in \{0, 1, \ldots, N\}$. Thus, meaningful estimation of the joint distribution $(X_1, X_2, Y)$ requires a batch of patient trajectories with sufficient label diversity; otherwise the joint distribution becomes degenerate.

We use MIMIC-IV IHM (in-hospital mortality) as a concrete example to study the importance of class balance and label quality for meaningful temporal RUS estimation. In-hospital mortality is a binary classification task where the label indicates whether the patient dies after ICU discharge, we designed three conditions to evaluate how class imbalance and label quality affect both RUS estimation and model performance:

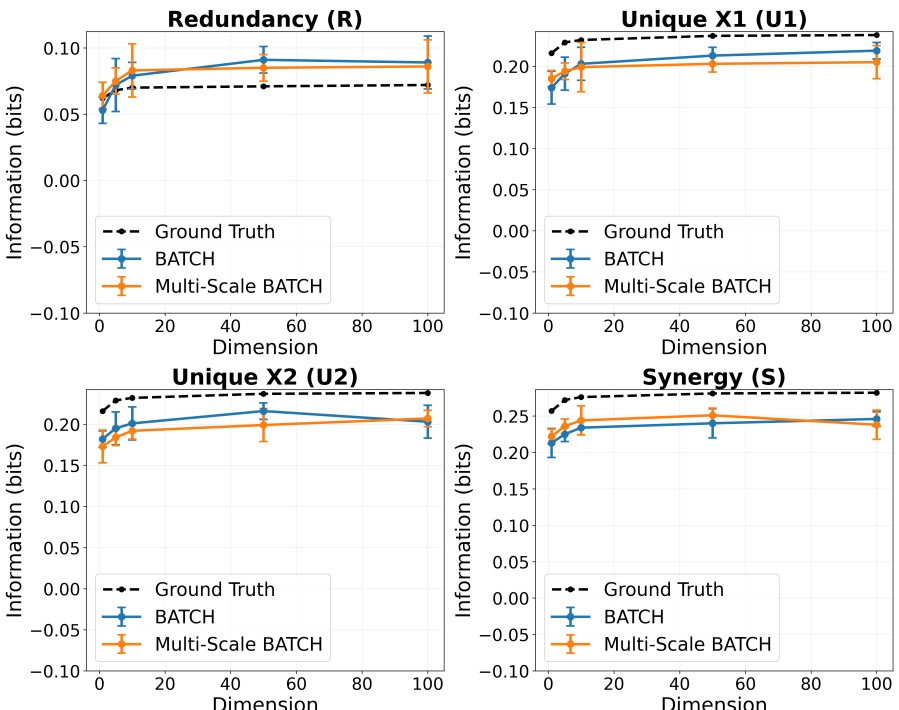

Figure 11: Ablation study on feature dimensionality. We evaluated the multi-scale BATCH estimator under varying feature dimensions $[1, 5, 10, 50, 100]$. Note that the ground-truth R/U/S values naturally change as the dimensionality of the features increases. Across all tested dimensions, both the multi-scale BATCH estimator and the step-wise computation produce consistent R/U/S estimates.

1. A batch containing only patients with outcome 0 or only outcome 1. In this case, temporal RUS cannot be meaningfully computed under the information decomposition framework because $(X_1, X_2, Y)$ becomes degenerate.

2. A batch containing both labels but with $> 95\%$ belonging to class 0. RUS can be computed, but the resulting temporal RUS sequence is substantially less informative, with the modality interaction over time remaining almost constant.

3. A batch with approximately balanced labels and sufficiently rich histories. This corresponds to the original temporal RUS used in the main experiments.

In Figure 16, we use the temporal RUS values of the "Insulin $\rightarrow$ Furosemide" modality pair as an illustrative example. Such results provide us insights on the importance of class balance and label quality for meaningful temporal RUS estimation, and demonstrate how degraded RUS estimation can affect the final model performance.

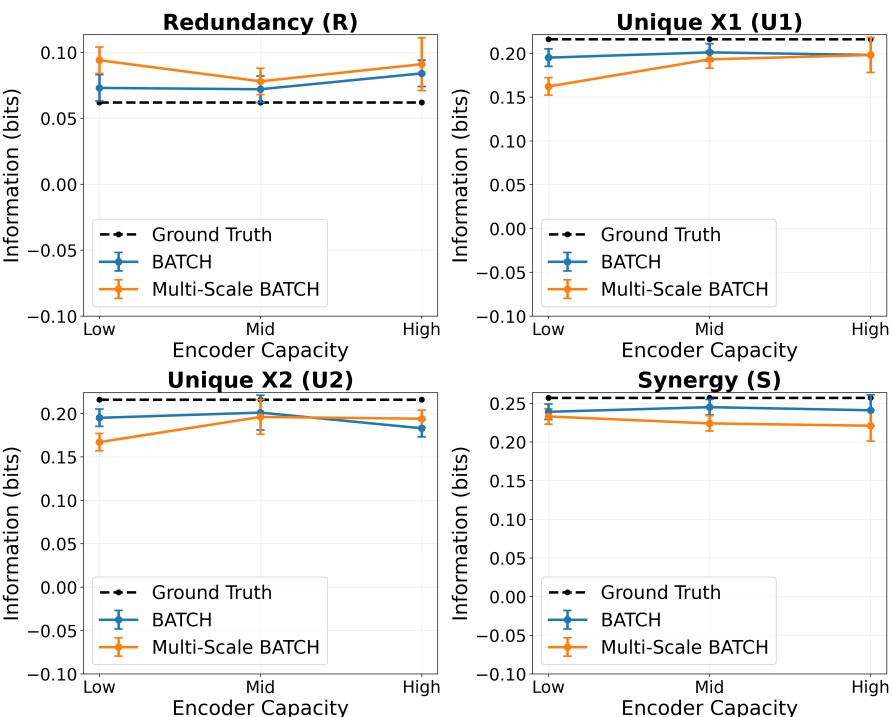

Figure 12: Ablation study on encoder capacity. We further tested the effect of encoder capacity in the BATCH estimator by evaluating 3 configurations: low, medium, and high capacity. In the low-capacity setting, the MLP encoder uses a hidden dimension of 16 with 1 layer; the medium-capacity setting uses a hidden dimension of 32 with 2 layers; and the high-capacity setting uses a hidden dimension of 64 with 3 layers. Across all tested scenarios, we observe that the low- and medium-capacity encoders are generally sufficient for accurate synthetic R/U/S estimation, while increasing the capacity yields only marginal improvements.

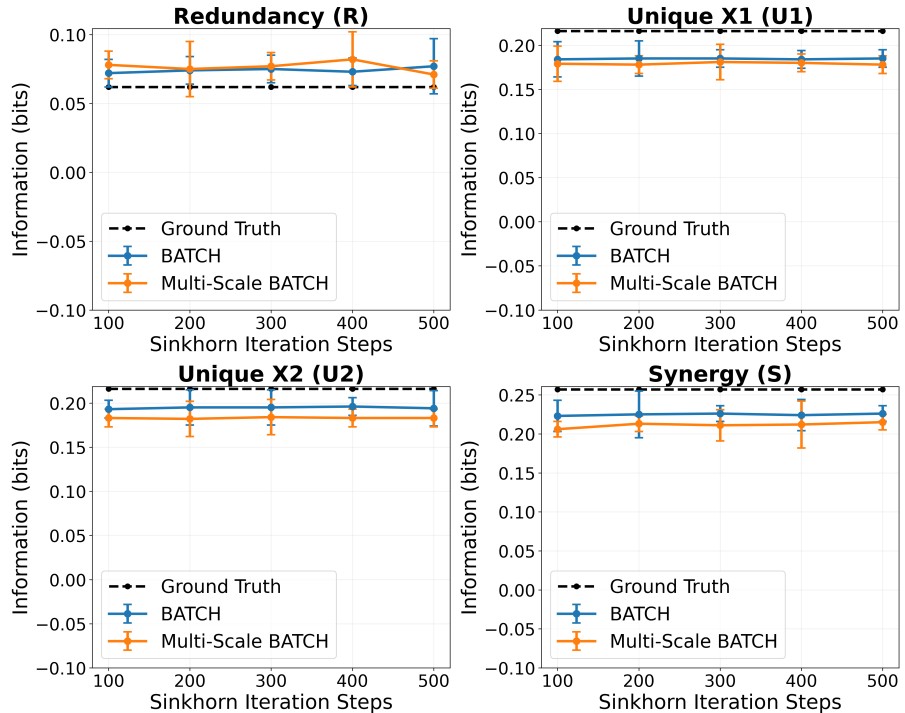

Figure 13: Ablation study on the number of Sinkhorn iteration steps. We evaluated both methods using a range of Sinkhorn iteration counts: $[100, 200, 300, 400, 500]$. Across all settings, the two methods produce close R/U/S estimates, and the estimator seems relatively insensitive to the choice of iteration count.

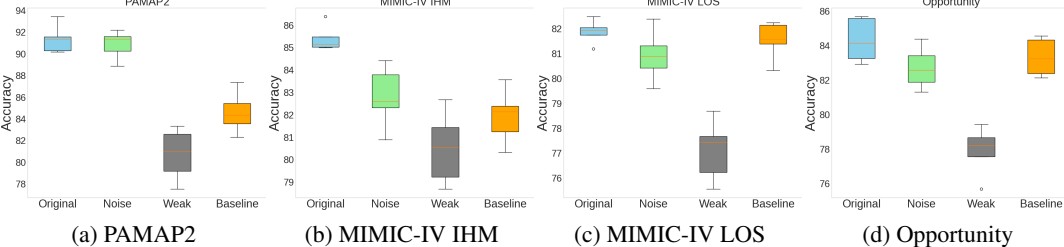

(a) PAMAP2   (b) MIMIC-IV IHM   (c) MIMIC-IV LOS   (d) Opportunity

Figure 14: Effect of inaccurate R/U/S estimation on model performance. Across the four evaluated datasets, we investigated how degrading the quality of the temporal R/U/S estimates impacts MERGE's performance. We observed that adding Gaussian noise to the R/U/S sequences has mild impact on downstream performance. However, making the R/U/S sequences weak or sparse leads to substantial performance degradation, often falling below baseline models. We speculate that noisy R/U/S values still preserve the overall structure of modality interactions, enabling the router to make approximately correct decisions. However, sparse or near-zero R/U/S sequences fail to provide meaningful interaction signals. As a result, the model receives misleading guidance and may adopt incorrect or inconsistent routing patterns.

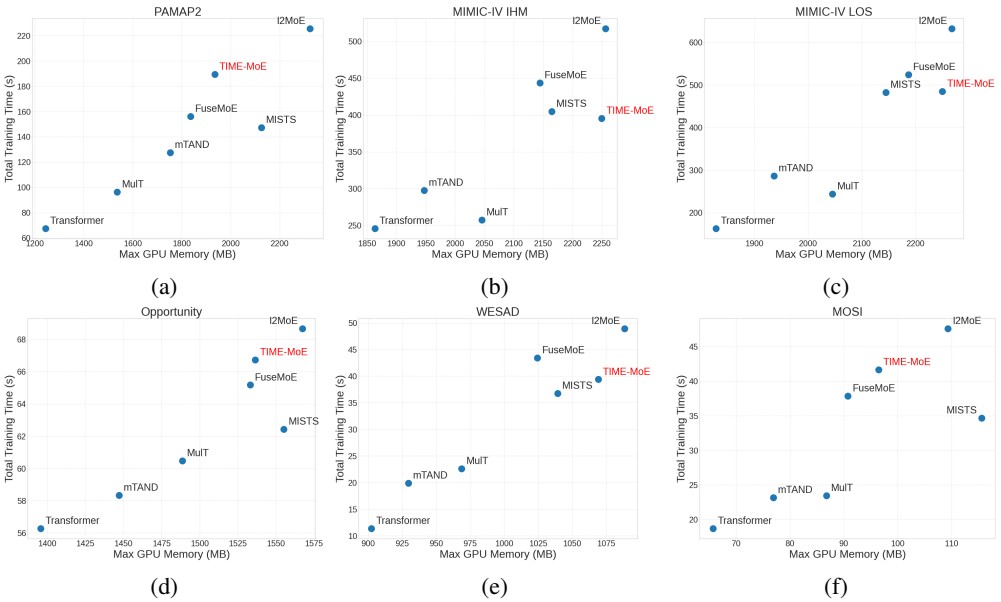

Figure 15: Comparison of training time and maximum GPU memory usage across all methods and datasets. Although computing and incorporating temporal RUS values into MoE training introduces a certain amount of additional computation and training time, the overall resource requirements of MERGE remain comparable to baseline multimodal Transformer/MoE architectures, such as FuseMoE (Han et al., 2024) and MISTS (Zhang et al., 2023), while being more efficient than I2MoE (Xin et al., 2025). Note that the sequence length of the temporal RUS values varies depending on the task and dataset; longer sequences naturally increase memory consumption. As a result, the relative positioning of MERGE in the training-time v.s. memory plot varies across datasets. Models such as the standard multi-head self-attention Transformer, mTAND, and MulT represent earlier-generation attention-based architectures with lower computational requirements, and thus appear on the lower end of the resource spectrum.

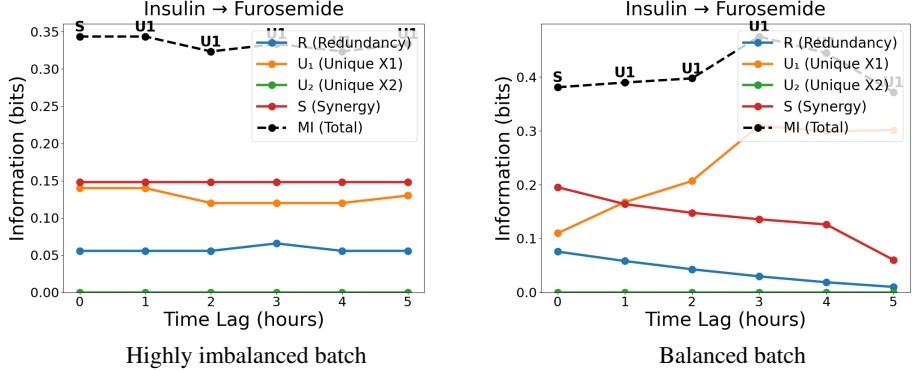

Figure 16: We use the "Insulin → Furosemide" modality pair as an example to compare temporal RUS estimations obtained under class-imbalanced settings versus those computed under the regular (balanced) setting. We fed RUS sequences from both *highly imbalanced batch* and *balanced batch* into MERGE training. We observed a clear performance degradation for the imbalanced batch: AUROC decreased from 85.40 to 82.35 and F1 score decreased from 84.97 to 81.73.

