# OpenReview forum: "Massively Multimodal Foundation Models: A Framework for Capturing Interactions with Specialized Mixture-of-Experts"
_ICLR.cc/2026/Conference — ICLR 2026 Poster_

### Official Review · Reviewer_sito · 2025-10-27

**Soundness:** 2
**Presentation:** 2
**Contribution:** 2
**Rating:** 6
**Confidence:** 2

**Summary:**

This paper presents TIME-MoE, a Temporal Interaction-guided Mixture-of-Experts framework that leverages information-theoretic decomposition (RUS) to guide expert routing.

**Strengths:**

The paper introduces a novel information-theoretic framework for guiding expert routing in multimodal MoE systems.

**Weaknesses:**

The proposed RUS estimation and routing involve substantial computational overhead and complex hyperparameter tuning. Scalability to large-scale multimodal models remains to be shown.

**Questions:**

1. The proposed multi-scale estimator is interesting, but the overall computation involving PID and Sinkhorn alignment still seems heavy, especially for long temporal sequences or high-dimensional inputs. It would be helpful if the authors could comment on the practical computational cost and scalability of TIME-MoE in such settings.

2. The model appears to rely on several threshold and weighting hyperparameters (τ_R, τ_U, τ_S, λ_R, λ_U, λ_S). How sensitive are the results to these choices? Any guidance or heuristics for tuning them would be useful.

3. Most of the experiments are on medium-scale datasets. Have the authors tried (or do they see a clear path to) applying TIME-MoE to larger-scale setups, e.g., vision-language or multimodal LLM tasks?

4. The framework is built on information-theoretic principles, but in practice RUS estimation depends on neural approximations. How stable are these estimations during training, and is there any quantitative sense of their bias or variance?

---

> ### Author Response · Authors · 2025-11-25
> **Response to Reviewer sito (Part 1)**
>
> Dear reviewer `sito`,
>
> We are grateful for your positive review and suggestions regarding our paper. Our response to your concern is detailed below, and we reference the linked additional results below to support our points. We hope this response addresses your questions.
>
> ### [Anonymous link to additional results.](https://sites.google.com/view/iclr26timemoe/home)
>
> **W1: The proposed RUS estimation and routing involve substantial computational overhead and complex hyperparameter tuning. Scalability to large-scale multimodal models remains to be shown.**
>
> **A:** We have added a comparison of the total training time (temporal RUS estimation + TIME-MoE training for our method) and the maximum GPU memory usage across all methods and datasets. Although computing and incorporating temporal RUS values into MoE training introduces a certain amount of additional computation and training time, the overall resource requirements of TIME-MoE remain comparable to baseline multimodal Transformer/MoE architectures, such as FuseMoE and MISTS, while being more efficient than I2MoE. The corresponding results are reported in **Experiment #3** in the link above.
>
> We would also like to clarify that the temporal RUS estimation is performed separately from the TIME-MoE training phase, but it uses the same training/validation/test partitions as the main model. This design choice does not introduce additional computational overhead. Since temporal RUS is a property of the dataset, it only needs to be computed once. The resulting values can then be cached and reused across any number of downstream tasks for this dataset.
>
> Furthermore, to demonstrate the effectiveness of TIME-MoE on a larger-scale multimodal generation task, we conducted an additional evaluation on the Ego4D activity caption generation task. The results for this experiment are provided in **Experiment #4** in the link above.
>
> **Q1: The proposed multi-scale estimator is interesting, but the overall computation involving PID and Sinkhorn alignment still seems heavy, especially for long temporal sequences or high-dimensional inputs. It would be helpful if the authors could comment on the practical computational cost and scalability of TIME-MoE in such settings.**
>
> **A:** In response to your request, we have added a comparison of the total training time and the maximum GPU memory usage across all methods and datasets under different configurations. The corresponding results are reported in **Experiment #3** as mentioned in our response to **W1**.
>
> **Q2: The model appears to rely on several threshold and weighting hyperparameters (τ_R, τ_U, τ_S, λ_R, λ_U, λ_S). How sensitive are the results to these choices? Any guidance or heuristics for tuning them would be useful.**
>
> **A:** The parameters $(\tau_R, \tau_U, \tau_S)$ serve as threshold values that determine when to activate different routing behaviors based on the RUS scores. Conceptually, they control the selectivity of the routing mechanism. If a threshold is set too low, almost all interactions will trigger specialized routing; if set too high, the routing reverts to a standard MoE regime and effectively ignores RUS-guided specialization. Empirically, synergistic interactions tend to be much rarer than redundancy or uniqueness in real-world multimodal data, so we typically set $\tau_S$ lower to avoid overly suppressing synergy-driven routing and to maintain a balanced distribution of routing patterns.
>
> The parameters $(\lambda_R, \lambda_U, \lambda_S)$ regulate the strength of the auxiliary loss terms corresponding to each interaction type. These can be selected based on domain knowledge, e.g., whether a dataset is dominated by synergistic patterns, highly independent modalities, etc., or adjusted to explicitly encourage certain forms of interaction. They can also be made adaptive during training, with validation performance guiding the adjustment.
>
> Although $\tau$ and $\lambda$ encode different aspects of the system, they are related in effect: increasing a threshold $\tau$ reduces how often a particular interaction pattern is activated, which is analogous to decreasing the corresponding $\lambda$, thereby weakening its influence in training.
>
> In practice, we find that because the empirical distribution of RUS values is highly non-uniform, the thresholds $\tau$ (typically set as specific percentiles of their respective RUS distributions) are relatively insensitive to fine-grained tuning. Therefore, we precompute and fix $\tau$ based on dataset statistics and only tune the $\lambda$ values during model training.

---

> ### Author Response · Authors · 2025-11-25
> **Response to Reviewer sito (Part 2)**
>
> **Q3: Most of the experiments are on medium-scale datasets. Have the authors tried (or do they see a clear path to) applying TIME-MoE to larger-scale setups, e.g., vision-language or multimodal LLM tasks?**
>
> **A:** We choose to start with foundational multimodal transformers for several reasons:
> - The temporal RUS values have a clean interpretation when modalities are discrete and well-defined, providing a clearer path for integration;
> - Demonstrating that temporal multimodal interaction improve routing on diverse benchmarks establishes the core principle before tackling LLM complexity;
> - Healthcare, activity recognition, and affective computing represent high-value domains where this approach already shows substantial gains.
>
> We would like to emphasize that the design philosophy of leveraging temporal multimodal interaction to guide expert routing is fundamentally sound and is readily scalable to LLM-MoE frameworks. For instance, incorporating multimodal inputs with known temporal RUS values can substantially improve fine-tuning of MoE-based LLMs by helping the model identify the appropriate experts, which is an aspect that remains a challenging problem in current research. However, applying our framework directly to state-of-the-art LLM-MoE architectures would require a few nontrivial architectural advancements which is another separate line of work we are conducting. Here we briefly summarize a few:
> - TIME-MOE routes tokens based on modality-level temporal RUS values computed for modality pairs (e.g., CXR ↔ clinical notes). In language model based MoEs, tokens within a single text sequence don't have explicit modality labels, the routing operates at a finer, semantically ambiguous level. Adapting temporal RUS would require defining meaningful "interaction units" within text rather than discrete modalities.
> - TIME-MOE dedicates synergy experts with cross-attention modules for high-synergy pairs. LLM-MoE experts are typically homogeneous FFNs. If we aim to preserve the original expert architecture, we must rethink how to incorporate synergy-aware routing without altering the expert structure itself.
>
> Given the above-mentioned context, and in light of the reviewers’ interest in applying TIME-MoE to larger-scale multimodal benchmarks, we have designed a new experiment that extends our current framework to a substantially larger multimodal generation setting: one that is closer in scale to LLM-based multimodal systems, while not yet requiring solutions to the architectural challenges discussed above. The results of this experiment are provided in **Experiment #4** as mentioned in our response to **W1**.
>
> This experiment illustrates a concrete example of our envisioned path for scaling up TIME-MoE in future work. In particular, beyond language-model–based MoE architectures, we believe the current TIME-MoE design shows even greater potential when applied to large-scale VLMs or world-models, where temporal multimodal interaction dynamics remain highly informative and can meaningfully improve model performance in impactful real-world applications.
>
> **Q4: The framework is built on information-theoretic principles, but in practice RUS estimation depends on neural approximations. How stable are these estimations during training, and is there any quantitative sense of their bias or variance?**
>
> **A:** We have added a synthetic benchmark that enables a thorough evaluation of the temporal R/U/S estimation step across several dimensions, quantifying the bias and variance of the estimation procedure. The results for this experiment are provided in **Experiment #1** in the link above. Furthermore, upon obtaining the temporal R/U/S estimations, we investigated the sensitivity of the final model's performance to the accuracy of the R/U/S estimation. Specifically, we observed that adding noise to the R/U/S sequences had a mild impact on downstream performance. However, making the R/U/S sequences weak or sparse led to performance degradation, often falling below baseline models. This indicates the critical importance of correctly leveraging the information carried by the temporal R/U/S sequence. Details of this investigation can be found in **Experiment #2** in the link above.
>
> Please do not hesitate to let us know if you have additional questions!
>
> Sincerely,
>
> Authors

---

### Official Review · Reviewer_z7XH · 2025-10-28

**Soundness:** 3
**Presentation:** 4
**Contribution:** 3
**Rating:** 8
**Confidence:** 4

**Summary:**

This paper introduces TIME-MoE (Temporal Interaction-guided Mixture of Experts), a novel multimodal architecture that integrates temporal interaction dynamics into the MoE routing process. The core idea is to quantify multimodal interactions (redundancy, uniqueness, synergy – RUS) over time, and use these dynamics to guide expert routing. The authors design a RUS-aware router and corresponding auxiliary losses that encourage experts to specialize in particular interaction types. Empirical results on multiple benchmarks (MIMIC-IV, PAMAP2, WESAD, MOSI, Opportunity) show that TIME-MoE outperforms state-of-the-art fusion and MoE-based baselines (e.g., FuseMoE, I2MoE), achieving both higher predictive accuracy and more interpretable expert activation patterns over temporal dimension.

**Strengths:**

1. Novel theoretical grounding: Extends Partial Information Decomposition (PID) to the temporal domain using directed information, leading to a measurable RUS metric that captures delayed effects.
2. Principled architecture: The RUS-aware router and auxiliary loss design directly encode theoretical insights into model training.
3. Empirical generality: Demonstrates improvements across heterogeneous datasets (clinical, affective, activity, physiological).
4. Interpretability: Expert routing visualizations reveal meaningful alignment with modality interactions (e.g., redundancy between chest and hand motion over the temporal dimension).
5. Comprehensive evaluation: Includes ablations, sensitivity to RUS sequence length, and comparison with multiple baselines.

**Weaknesses:**

W1. Computing temporal RUS and training the router could be computationally expensive, which may limit scalability for large multimodal models.

**Questions:**

1. How sensitive is TIME-MoE to errors in RUS estimation, especially when temporal dependencies are weak or noisy?
2. Can the RUS-guided router be learned end-to-end without explicit precomputation of RUS values?
3. Could the same framework generalize to non-temporal multimodal fusion (e.g., static image–text tasks) or to LLM-scale architectures?
4. What is the computational overhead compared to FuseMoE and I2MoE in FLOPs and training time?
5. How does the proposed directed-information-based temporal RUS compare with simpler correlation-based interaction metrics?

---

> ### Author Response · Authors · 2025-11-25
> **Response to Reviewer z7XH (Part 1)**
>
> Dear reviewer `z7XH`,
>
> We sincerely appreciate your positive review and suggestions for our paper's improvement. Our detailed response to your concern is below, and we would like to point you to a link for additional results, which will serve as a reference in this response. We hope our response addresses your questions.
>
> ### [Anonymous link to additional results.](https://sites.google.com/view/iclr26timemoe/home)
>
> **W1. Computing temporal RUS and training the router could be computationally expensive, which may limit scalability for large multimodal models.**
>
> **A:** We have added a comparison of the total training time (temporal RUS estimation + TIME-MoE training for our method) and the maximum GPU memory usage across all methods and datasets. The corresponding results are reported in **Experiment #3** in the link above.
>
> We would also like to clarify that the temporal RUS estimation is performed separately from the TIME-MoE training phase, but it uses the same training/validation/test partitions as the main model. This design choice does not introduce additional computational overhead. Since temporal RUS is a property of the dataset, it only needs to be computed once. The resulting values can then be cached and reused across any number of downstream tasks for this dataset.
>
> **Q1: How sensitive is TIME-MoE to errors in RUS estimation, especially when temporal dependencies are weak or noisy?**
>
> **A:** In response to your request, we have added a detailed ablation study to investigate the effect of bad R/U/S estimations on the model performance. We observed that adding noise to the R/U/S sequences has a mild impact on downstream performance. However, making the R/U/S sequences weak or sparse leads to performance degradation, often falling below baseline models. Details of this experiment can be found in **Experiment #2** in the link above.
>
> **Q2: Can the RUS-guided router be learned end-to-end without explicit precomputation of RUS values?**
>
> **A:** Thank you for the insightful comment. While it is indeed possible to build an end-to-end framework that jointly learns temporal RUS values and MoE model, we intentionally decouple these two components for both methodological and practical reasons.
>
> First, the multi-scale batch estimator used for computing temporal RUS sequences is designed to reflect intrinsic information-theoretic structure of the dataset. It should not be optimized to minimize task-specific losses, doing so would entangle the RUS values with downstream objectives, undermining their role as task-agnostic signals of temporal multimodal interactions. Training the estimator separately ensures that the resulting RUS values remain invariant across tasks and thus maintain their generalization capacity.
>
> Second, this design choice does not introduce additional computational overhead. Since temporal RUS is a property of the dataset, it only needs to be computed once. The resulting values can then be cached and reused across any number of downstream tasks for this dataset. In contrast, integrating RUS estimation directly into end-to-end model training would (1) increase training time, and (2) introduce optimization instability.
>
> **Q3: Could the same framework generalize to non-temporal multimodal fusion (e.g., static image–text tasks) or to LLM-scale architectures?**
>
> **A:** The current framework can naturally be applied to non-temporal multimodal interaction. In fact, temporal multimodal interaction is a more general formulation: static interactions can simply be represented as RUS sequences with constant values. As stated early in our draft (around line 39), prior methods typically capture only static interactions and ignore temporal structure; our method extends this by modeling temporal dynamics and using them to guide MoE routing. While there is also no prior work that uses static interactions to inform MoE routing, we believe addressing the more general and challenging temporal case provides greater value to the community.
>
> Additionally, given the reviewers’ interest in applying TIME-MoE to larger-scale multimodal benchmarks, we designed a new experiment that adapts our framework to a substantially larger multimodal generation setup, one that is closer in scale to LLM-based multimodal systems. The results of this experiment are provided in **Experiment #4** in the link above.
>
> **Q4: What is the computational overhead compared to FuseMoE and I2MoE in FLOPs and training time?**
>
> **A:** In response to your request, we have added a comparison of the total training time (temporal RUS estimation + TIME-MoE training for our method) and the maximum GPU memory usage across all methods and datasets. The corresponding results are reported in **Experiment #3** in the link above.

---

> ### Author Response · Authors · 2025-11-25
> **Response to Reviewer z7XH (Part 2)**
>
> **Q5: How does the proposed directed-information-based temporal RUS compare with simpler correlation-based interaction metrics?**
>
> **A:** Thank you for the question. Since the review did not specify a particular correlation-based metric, we evaluated the most commonly used options and their extensions. Below we list several fundamental limitations in the temporal and multimodal settings we study.
>
> 1. Take Pearson correlation as an example, such a metric has no sense of time: it cannot distinguish whether $X_1(t-2)$ influences $Y(t)$ or whether $Y(t)$ influences $X_1(t-2)$. Even with sliding-windows, correlation remains a symmetric measure and therefore cannot capture directionality.
>
> 2. Most correlation metrics are also restricted to linear relationships, missing key nonlinear or delayed interactions that are important in real applications.
>
> 3. Moreover, they can only reveal whether a signal is similar or unique relative to others. They cannot distinguish redundancy from synergy. Correlation-based methods ultimately reduce interactions to a single aggregated score. In contrast, our directed-information–based temporal RUS framework decomposes information flow across time lags, yielding a rich temporal profile of how modalities influence each other.
>
> 4. Most importantly, correlation may group modalities together simply because they are highly correlated: even when the correlation is spurious or non-causal, leading to suboptimal expert specialization. In contrast, our RUS-guided routing is based on principled distinctions between redundancy, uniqueness, and synergy, which enables more meaningful expert assignments.
>
> Please do not hesitate to let us know if you have additional questions!
>
> Sincerely,
>
> Authors

---

### Official Review · Reviewer_RmCG · 2025-11-01

**Soundness:** 3
**Presentation:** 3
**Contribution:** 3
**Rating:** 4
**Confidence:** 3

**Summary:**

This paper presents the study on Temporal Interaction-guided Mixture-of-Experts (TIME-MoE). This is a modified MoE architecture that explicitly leverages time-varying multimodal interactions to guide expert routing. In conventional MoE routers the tokens are navigated based solely on content similarity, while for some tasks and inputs, the temporal interaction between tokens is crucial. TIME-MoE's router is interaction-aware, it considers the redundancy, uniqueness, and synergy (RUS) between modalities over time when deciding which expert processes which token. By using quantified interaction dynamics as a prior, the model encourages experts to specialize in processing particular interaction patterns (redundant, unique, or synergistic information) rather than only learning task-specific features.
The authors testes their methods on various multimodal benchmarks, mostly connected to the medical domain, and found out that their method outperforms alternative MoEs approaches, and also improves interpretability of the method.

**Strengths:**

The main strengths of the research are the following:

1. Novel interaction-aware MoE for multimodal data. The authors integrate an information-theoretic framework with modern deep learning to guide Mixture-of-Experts routing using temporal redundancy, uniqueness, and synergy (RUS). This design yields an MoE that is both temporally and modality-grounded, leading to more specialized and interpretable experts.
2. Unlike prior approaches that rely on static cross-modal correlations, TIME-MoE explicitly models time-lagged multimodal effects, which are especially important in medical and physiological domains.
3. Thorough and diverse evaluation. The approach is tested on a broad set of multimodal benchmarks — PAMAP2, MIMIC-IV (IHM/LOS), MOSI, WESAD, and Opportunity — covering various modality mix. The authors validate their mechanism through component ablations and routing-pattern analyses, confirming both effectiveness and interpretability.
4. Comprehensive ablation studies. Each component (redundancy, uniqueness, synergy guidance) is evaluated separately, demonstrating its distinct contribution to the final performance.

**Weaknesses:**

I would define the following weaknesses:

1. Pairwise-only interaction modeling. The RUS framework focuses on pairwise modality interactions; higher-order (three-way or more) synergies are not explicitly modeled. While this ensures scalability, it limits the expressiveness of the model.
2. Although the authors claim a τ-fold speedup in RUS computation and improved parameter efficiency, the paper does not provide GPU hours, or memory-usage comparisons. Including such measurements would make the efficiency claims more convincing.
3. Integration with large-scale LLM-MoE frameworks remains unexplored. Given the rapid development of Large Language Model MoEs (e.g., Mixtral, DeepSeek-MoE, etc.), it would be highly valuable to discuss how TIME-MoE could scale to these architectures. The paper does not explore or speculate on integrating its RUS-guided routing into LLM-scale systems, which could significantly broaden its impact.
4. While Appendix E describes architecture and hyperparameters, and Appendix D details the RUS estimation algorithms, the paper lacks a concise description of the training setup (optimizer, learning rate, batch size, epochs, etc.).

**Questions:**

1. Please clarify the optimizer, learning-rate schedule, batch size, number of epochs, and gating top-k used during TIME-MoE training. Were modality encoders frozen or fine-tuned?
2. Is RUS estimation performed as a fully offline pre-processing step (aligned with train/validation/test splits), or is there any joint optimization of the RUS estimator with TIME-MoE training?
3. Did you observe any expert collapse (e.g., one expert processing the majority of tokens)?
4. How sensitive is the final model’s performance to the accuracy of the RUS estimation? If the interaction estimator produces noisy or weak signals, does performance degrade significantly?
5. Could you report wall-clock training times, GPU type/count, and memory usage for both RUS estimation and TIME-MoE training, relative to baseline MoE models (e.g., FuseMoE, I2MoE)?
6. Do you plan to extend TIME-MoE to large-language-model MoE architectures? Such integration could yield interpretable and context-aware routing in large-scale generative models.

If the authors address my questions, I am willing to increase the final score.

---

> ### Author Response · Authors · 2025-11-25
> **Response to Reviewer RmCG (Part 1)**
>
> Dear reviewer `RmCG`:
>
> We sincerely appreciate your constructive suggestions, which have greatly helped us improve the paper. Our detailed responses to your concerns are provided below, and we hope they can address your questions.
>
> **W1: Pairwise-only interaction modeling. The RUS framework focuses on pairwise modality interactions; higher-order (three-way or more) synergies are not explicitly modeled. While this ensures scalability, it limits the expressiveness of the model.**
>
> **A:** Thank you for the insightful comments. At the early stage of developing the mathematical foundation for this work, we did consider extending the framework to higher-order interactions, as discussed in [1], with the goal of uncovering more nuanced discrepancies in the underlying multimodal data-generating mechanisms, such as chain, v-structure, and fork patterns in the implied causal graph formed by multimodal variables. We ultimately decided not to pursue this direction in the current work for several reasons:
>
> (1) Mismatch with MoE routing granularity. MoE routing is token-wise and fundamentally similarity-based. A key contribution of our work is incorporating pairwise (and temporal) modality relationships directly into the routing mechanism. Once the routing framework is anchored on pairwise interactions, higher-order interactions such as three-way synergy do not translate cleanly into actionable routing decisions. For example, if a meaningful interaction only emerges when modalities A, B, and C are jointly present, it is unclear how to map this three-way dependency into expert assignments that operate strictly on pairwise token-level relationships. Moreover, such higher-order interactions require the simultaneous presence of all three modalities, which occurs far less frequently in practice than two-modality combinations.
>
> (2) In domains such as activity recognition, practitioners typically care about understanding which modalities share information for efficient processing rather than inferring more intricate higher-order causal chains (e.g., chest → hand → ankle). Pairwise interaction patterns tend to offer clearer, more actionable insights for model design, feature engineering, and interpretability.
>
> (3) Across the datasets we evaluate, most meaningful multimodal dynamics are well captured at the pairwise level. For instance, clinical monitoring often focuses on direct physiological relationships (e.g., heart rate ↔ blood pressure), and higher-order interactions provide diminishing returns relative to their computational cost.
>
> Given these considerations, we believe that starting with pairwise temporal interaction modeling is both effective and aligned with MoE’s characteristics. Nonetheless, we fully agree that higher-order multimodal interactions may have their own value, particularly for causal discovery and deeper multimodal understanding. Extending our framework to accommodate such interactions can be explored in follow-up research.
>
> **Reference**
>
> [1] Ghassami et al. (2017), Interaction Information for Causal Inference: The Case of Directed Triangle.
>
> **W2: Although the authors claim a τ-fold speedup in RUS computation and improved parameter efficiency, the paper does not provide GPU hours, or memory-usage comparisons. Including such measurements would make the efficiency claims more convincing.**
>
> **A:** In response to your request, we have added a comparison of the total training time (temporal RUS estimation + TIME-MoE training for our method) and the maximum GPU memory usage across all methods and datasets. The corresponding results are reported in **Experiment #3** in the link below. In addition, concerning the effectiveness of the $\tau$-fold speedup in RUS computation, we have also added a synthetic benchmark that enables a thorough evaluation of our multi-scale BATCH estimator across several dimensions, including sample sizes, feature dimensions, encoder capacities, and numbers of Sinkhorn iterations. The results for this experiment are provided in **Experiment #1** in the same link.
>
> ### [Anonymous link to additional results.](https://sites.google.com/view/iclr26timemoe/home)

---

> ### Author Response · Authors · 2025-11-25
> **Response to Reviewer RmCG (Part 2)**
>
> **W3: Integration with large-scale LLM-MoE frameworks remains unexplored. Given the rapid development of Large Language Model MoEs (e.g., Mixtral, DeepSeek-MoE, etc.), it would be highly valuable to discuss how TIME-MoE could scale to these architectures. The paper does not explore or speculate on integrating its RUS-guided routing into LLM-scale systems, which could significantly broaden its impact.**
>
> **A:** Thank you for your insightful suggestion. We choose to start with foundational multimodal transformers for several reasons:
> - The temporal RUS values have a clean interpretation when modalities are discrete and well-defined, providing a clearer path for integration;
> - Demonstrating that temporal multimodal interaction improve routing on diverse benchmarks establishes the core principle before tackling LLM complexity;
> - Healthcare, activity recognition, and affective computing represent high-value domains where this approach already shows substantial gains.
>
> We would like to emphasize that the design philosophy of leveraging temporal multimodal interaction to guide expert routing is fundamentally sound and is readily scalable to LLM-MoE frameworks. For instance, incorporating multimodal inputs with known temporal RUS values can substantially improve fine-tuning of MoE-based LLMs by helping the model identify the appropriate experts, which is an aspect that remains a challenging problem in current research. However, applying our framework directly to state-of-the-art LLM-MoE architectures would require a few nontrivial architectural advancements which is another separate line of work we are conducting. Here we briefly summarize a few:
> - TIME-MOE routes tokens based on modality-level temporal RUS values computed for modality pairs (e.g., CXR ↔ clinical notes). In language model based MoEs, tokens within a single text sequence don't have explicit modality labels, the routing operates at a finer, semantically ambiguous level. Adapting temporal RUS would require defining meaningful "interaction units" within text rather than discrete modalities.
> - TIME-MOE dedicates synergy experts with cross-attention modules for high-synergy pairs. LLM-MoE experts are typically homogeneous FFNs. If we aim to preserve the original expert architecture, we must rethink how to incorporate synergy-aware routing without altering the expert structure itself.
>
> Given the above-mentioned context, and in light of the reviewers’ interest in applying TIME-MoE to larger-scale multimodal benchmarks, we have designed a new experiment that extends our current framework to a substantially larger multimodal generation setting: one that is closer in scale to LLM-based multimodal systems, while not yet requiring solutions to the architectural challenges discussed above. The results of this experiment are provided in **Experiment #4** in the link below.
>
> ### [Anonymous link to additional results.](https://sites.google.com/view/iclr26timemoe/home)
>
> This experiment illustrates a concrete example of our envisioned path for scaling up TIME-MoE in future work. In particular, beyond language-model–based MoE architectures, we believe the current TIME-MoE design shows even greater potential when applied to large-scale VLMs or world-models, where temporal multimodal interaction dynamics remain highly informative and can meaningfully improve model performance in impactful real-world applications.
>
> **W4: While Appendix E describes architecture and hyperparameters, and Appendix D details the RUS estimation algorithms, the paper lacks a concise description of the training setup (optimizer, learning rate, batch size, epochs, etc.).**
>
> **A:** In response to your request, we have added a concise description of the training setup for all datasets presented in the main paper. These details are summarized in tabular form and can be found in **Experiment #5** in the link provided below.
>
> ### [Anonymous link to additional results.](https://sites.google.com/view/iclr26timemoe/home)

---

> ### Author Response · Authors · 2025-11-25
> **Response to Reviewer RmCG (Part 3)**
>
> **Q1: Please clarify the optimizer, learning-rate schedule, batch size, number of epochs, and gating top-k used during TIME-MoE training. Were modality encoders frozen or fine-tuned? 2. Is RUS estimation performed as a fully offline pre-processing step (aligned with train/validation/test splits), or is there any joint optimization of the RUS estimator with TIME-MoE training?**
>
> **A:** (1) Please refer to **W4** for our detailed answer regarding the training setup. For the larger-scale MIMIC-IV benchmarks, all multimodal inputs are first encoded using pre-trained modality encoders, which are then fine-tuned jointly during TIME-MoE training. For the remaining datasets, the modality encoders are trained from scratch alongside TIME-MoE.
>
> (2) The RUS estimation is performed separately from the TIME-MoE training phase, but it uses the same training/validation/test partitions as the main model. While it is possible to build an end-to-end framework that jointly learns temporal RUS values and MoE model, we intentionally decouple these two components for both methodological and practical reasons.
>
> First, the multi-scale batch estimator used for computing temporal RUS sequences is designed to reflect intrinsic information-theoretic structure of the dataset. It should not be optimized to minimize task-specific losses, doing so would entangle the RUS values with downstream objectives, undermining their role as task-agnostic signals of temporal multimodal interactions. Training the estimator separately ensures that the resulting RUS values remain invariant across tasks and thus maintain their generalization capacity.
>
> Second, this design choice does not introduce additional computational overhead. Since temporal RUS is a property of the dataset, it only needs to be computed once. The resulting values can then be cached and reused across any number of downstream tasks for this dataset. In contrast, integrating RUS estimation directly into end-to-end model training would (1) increase training time, and (2) introduce optimization instability.
>
> **Q2: Did you observe any expert collapse (e.g., one expert processing the majority of tokens)?**
>
> **A:** We did observe expert collapse in early experiments on the MOSI dataset. With the initial auxiliary loss weights set to $\lambda_R = \lambda_U = \lambda_S = 2$, nearly all tokens were routed to the two synergy experts, leaving the remaining redundancy and uniqueness experts unused. We suspected that this is due to the overemphasis on the synergy loss that enforces all tokens to be routed to the synergy experts. After tuning these auxiliary loss weights, we identified a more stable configuration ($\lambda_R = 1,\ \lambda_U = 0.5,\ \lambda_S = 0.1$) that results in full utilization of all experts. Under this setting, the two synergy experts primarily process vision and text tokens jointly, while the remaining experts focus predominantly on audio tokens. This configuration also yields better performance than the collapsed setting.
>
> We did not observe expert collapse for any of the other datasets. We kindly refer you to the response **Q2** to reviewer `sito`, for what these parameters indicate and how to tune the parameters to achieve better performance on multimodal applications.
>
> **Q3: How sensitive is the final model’s performance to the accuracy of the RUS estimation? If the interaction estimator produces noisy or weak signals, does performance degrade significantly?**
>
> **A:** In response to your request, we have added a detailed ablation study to investigate the effect of bad R/U/S estimations on the model performance. We observed that adding noise to the R/U/S sequences has a mild impact on downstream performance. However, making the R/U/S sequences weak or sparse leads to performance degradation, often falling below baseline models. Details of this experiment can be found in **Experiment #2** in the link provided below.
>
> ### [Anonymous link to additional results.](https://sites.google.com/view/iclr26timemoe/home)
>
> **Q4: Could you report wall-clock training times, GPU type/count, and memory usage for both RUS estimation and TIME-MoE training, relative to baseline MoE models (e.g., FuseMoE, I2MoE)?**
>
> **A:** Please refer to **W2** for our detailed response to this question.
>
> **Q5: Do you plan to extend TIME-MoE to large-language-model MoE architectures? Such integration could yield interpretable and context-aware routing in large-scale generative models.**
>
> **A:** Please refer to **W3** for our detailed response to this question.
>
> Please do not hesitate to let us know if you have additional questions!
>
> Sincerely,
>
> Authors

---

> ### Author Response · Authors · 2025-11-28
> **Follow-Up Regarding Our Response to Your Review**
>
> Dear Reviewer `RmCG`,
>
> We hope this message finds you well. As the discussion period is progressing, we would like to kindly follow up regarding our response to your review. If there are any additional questions, concerns, or points you would like us to clarify, please let us know. Your insights are invaluable to us, and we are happy to clarify any remaining points that could help with your evaluation.
>
> We sincerely appreciate the time and effort you have dedicated to evaluating our paper and would be very grateful for any further comments you may have during the discussion phase.
>
> Thank you again for your thoughtful feedback and for reviewing our submission.
>
> Sincerely,
>
> Authors

---

### Official Review · Reviewer_UcoF · 2025-11-01

**Soundness:** 3
**Presentation:** 3
**Contribution:** 3
**Rating:** 8
**Confidence:** 3

**Summary:**

This work introduces TIME-MOE, which routes tokens in a mixture-of-experts using time-aware interaction cues between modalities. The model estimates redundancy, uniqueness, and synergy (RUS) via a directed-information flavored partial information decomposition, producing sequences over different lags. These sequences condition the router, while auxiliary losses encourage consistent behavior: co-route redundant inputs, diversify for unique inputs, and prefer dedicated synergy experts when cross-modal effects appear.

**Strengths:**

1)	Temporal interaction modeling: The paper builds on PID with directed information to model time-lagged redundancy, uniqueness, and synergy between modalities. The derivation reads cleanly and fits the needs of multimodal time series.
2)	Architecture: The RUS-aware router plus synergy experts turn the interaction signals into concrete routing decisions. The activation patterns look modality-consistent and more structured than a standard MoE.
3)	Efficient multi-lag estimation: The multi-scale estimator reuses computation across lags, staying close to step-wise estimates while delivering roughly τ-fold speedups.
4)	Empirical results: Strong empirical results & breadth: TIME-MOE wins on most metrics across 6 tasks, including clinical (MIMIC-IV) and affective datasets (MOSI, WESAD); results averaged over 5 runs.
5)	Reproducibility: Anonymous code link and detailed appendices and hyper params are provided.

**Weaknesses:**

1)	Estimator guarantees and identifiability: The construction of optimal Q*_{tau} via Sinkhorn-normalized alignment tensors is elegant but lacks formal guarantees about convergence to the PID-consistent minimizer and the induced bias in R/U/S under finite data and high-dimensional encoders. A short theorem or calibration study (e.g., on synthetic systems with known RUS) would make the claims much stronger.
2)	Dependency on label availability and distribution: For sequence-level labels (MIMIC-IV, MOSI), RUS is computed at a single time step while sweeping lags from the sequence end. That’s reasonable, but it could be brittle under class imbalance or label noise. Please analyze sensitivity and discuss how label quality affects RUS and routing.
3)	Scalability and cost: There is qualitative speedup from the multi-scale RUS estimator, but the paper could better quantify end-to-end training cost (GPU-hours) and router overhead vs. a standard MoE for larger expert counts and more modalities.

**Questions:**

1)	Estimator validation: Could You please provide a synthetic benchmark where ground-truth RUS over lags is known, to calibrate the multi-scale estimator vs. a step-wise computation, including error bars across data sizes? How sensitive are RUS estimates to encoder capacity and the Sinkhorn regularization?
2)	Label choice: For MIMIC-IV and MOSI, have you tried alternative targets (e.g., intermediate pseudo-labels or weak supervision) to compute more informative temporal RUS sequences? Any performance deltas?
3)	Computationals : Please report GPU-hours for TIME-MOE vs. other MoE baselines across datasets, and the incremental cost attributable specifically to RUS estimation and auxiliary losses.

---

> ### Author Response · Authors · 2025-11-24
> **Response to Reviewer UcoF (Part 1)**
>
> Dear reviewer `UcoF`:
>
> We sincerely appreciate your positive review and suggestions for our paper's improvement. Our response to your concern is detailed below, and we reference the linked additional results below to support our points. We hope this response addresses your questions.
>
> ### [Anonymous link to additional results.](https://sites.google.com/view/iclr26timemoe/home)
>
> **W1: Estimator guarantees and identifiability: The construction of optimal $Q_{\tau}$ via Sinkhorn-normalized alignment tensors is elegant but lacks formal guarantees about convergence to the PID-consistent minimizer and the induced bias in R/U/S under finite data and high-dimensional encoders. A short theorem or calibration study (e.g., on synthetic systems with known RUS) would make the claims much stronger.**
>
> **A:** In response to your request, we have incorporated a synthetic benchmark that enables a thorough evaluation of our multi-scale BATCH estimator framework across several dimensions. Detailed methodology and results can be found in **Experiment #1** in the link above.
>
> **W2: Dependency on label availability and distribution: For sequence-level labels (MIMIC-IV, MOSI), RUS is computed at a single time step while sweeping lags from the sequence end. That’s reasonable, but it could be brittle under class imbalance or label noise. Please analyze sensitivity and discuss how label quality affects RUS and routing.**
>
> **A:** Thank you for raising this important point. For datasets with sequence-level labels such as MIMIC-IV and MOSI, the prediction task is performed at the full-sequence level. For example, in MIMIC-IV IHM, the model uses the first 48 hours of ICU measurements to predict in-hospital mortality. Accordingly, we compute temporal RUS at the final time step (i.e., sweeping lags backward from the end of the 48-hour window), since these values are the relevant values aligned with the prediction target.
>
> We agree that class imbalance and label noise can influence both temporal RUS estimation and downstream routing behavior. In fact, these factors can affect any supervised signal used to compute relevance, including timestep-level variants. As noted in Section 4.1 of the main paper, each user/patient contains multiple measurements $\mathcal{D}_i \in \mathbb{R}^{d \times T}$, but is associated with only a single label $Y_i \in \{0, 1, \dots, N\}$. Thus, meaningful estimation of the joint distribution $(X_1, X_2, Y)$ requires a batch of user/patient trajectories with sufficient label diversity; otherwise the joint distribution becomes degenerate.
>
> Using MIMIC-IV IHM as a concrete example: a binary classification task where the label indicates whether the patient dies after ICU discharge, we designed three conditions to evaluate how class imbalance and label quality affect both RUS estimation and model performance:
>
> (1) A batch containing only patients with outcome 0 or only outcome 1. In this case, temporal RUS cannot be meaningfully computed under the information decomposition framework because $(X_1, X_2, Y)$ becomes degenerate.
>
> (2) A batch containing both labels but with >95% belonging to class 0. RUS can be computed, but the resulting temporal RUS sequence is substantially less informative, with the modality interaction over time remaining almost constant.
>
> (3) A batch with approximately balanced labels and sufficiently rich histories. This corresponds to the original temporal RUS used in the main experiments.
>
> As shown in **Experiment #6** in the link above, we use the temporal RUS values of the “Insulin → Furosemide’’ modality pair as an illustrative example. We then fed these RUS sequences from conditions (2) and (3) into TIME-MoE training. We observed a clear performance degradation for condition (2): AUROC decreased from 85.40 → 82.35 and F1 score decreased from 84.97 → 81.73. Such results provide us insights on the importance of class balance and label quality for meaningful temporal RUS estimation, and demonstrate how degraded RUS estimation can affect the final model performance.

---

> ### Author Response · Authors · 2025-11-26
> **Response to Reviewer UcoF (Part 2)**
>
> **W3: Scalability and cost: There is qualitative speedup from the multi-scale RUS estimator, but the paper could better quantify end-to-end training cost (GPU-hours) and router overhead vs. a standard MoE for larger expert counts and more modalities.**
>
> **A:** In response to your request, we have added a comparison of the total training time (temporal RUS estimation + TIME-MoE training for our method) and the maximum GPU memory usage across all methods and datasets under different configurations. The corresponding results are reported in **Experiment #3** in the link above. Furthermore, to demonstrate the effectiveness of TIME-MoE on a larger-scale multimodal generation task, we conducted an additional evaluation on the Ego4D activity caption generation task. The results for this experiment are provided in **Experiment #4** in the same link.
>
> **Q1: Estimator validation: Could You please provide a synthetic benchmark where ground-truth RUS over lags is known, to calibrate the multi-scale estimator vs. a step-wise computation, including error bars across data sizes? How sensitive are RUS estimates to encoder capacity and the Sinkhorn regularization?**
>
> **A:** Please see our response in **W1**.
>
> **Q2: Label choice: For MIMIC-IV and MOSI, have you tried alternative targets (e.g., intermediate pseudo-labels or weak supervision) to compute more informative temporal RUS sequences? Any performance deltas?**
>
> **A:** Please refer to our response to **W2** for our study on the importance of class balance and label quality for meaningful temporal RUS estimation. As noted earlier, both MIMIC-IV and MOSI involve full-sequence classification. Therefore, we compute temporal RUS using the sequence-level ground-truth label, ensuring that RUS values are aligned with how the model is ultimately used.
>
> Introducing pseudo-labels or weak supervision for intermediate timesteps could indeed be a viable option for timestep-level classification settings. However, for sequence-level tasks, such an approach would likely introduce a mismatch between the labels used during RUS computation and the labels used during model training and evaluation. This discrepancy can distort the underlying input–label relationship and degrade both RUS estimation quality and final model performance.
>
> **Q3: Computationals: Please report GPU-hours for TIME-MOE vs. other MoE baselines across datasets, and the incremental cost attributable specifically to RUS estimation and auxiliary losses.**
>
> **A:** Please see our response in **W3**.
>
> Please do not hesitate to let us know if you have additional questions!
>
> Sincerely,
>
> Authors

---

### Author Response · Authors · 2025-11-24
**Summary of Strengths and Updates on New Results**

Dear Area Chair and Reviewers,

We sincerely thank you for your valuable feedback and constructive reviews, which have greatly contributed to improving our paper. The following endorsements are truly encouraging:

- Novelty: Our paper introduces a novel Temporal Interaction-guided Mixture-of-Experts architecture that explicitly leverages time-varying multimodal interactions for expert routing (Reviewers `RmCG`, `z7XH`, `UcoF`, `sito`).
- Theory and Design: Reviewers praised the principled theoretical grounding: extending Partial Information Decomposition to the temporal domain using directed information, and its clear integration into the router and auxiliary losses (Reviewers `z7XH`, `UcoF`).
- Architecture and Interpretability: The RUS-aware router and synergy experts were noted to produce structured and interpretable expert activations aligned with modality interactions (Reviewers `RmCG`, `UcoF`, `z7XH`).
- Empirical Evaluation: The paper demonstrates strong empirical performance and breadth, outperforming state-of-the-art baselines across diverse multimodal benchmarks (Reviewers `RmCG`, `z7XH`, `UcoF`).
- Ablations and Analysis: The comprehensive ablation studies and sensitivity analyses validate each design component and confirm the interpretability of routing behavior (Reviewers `RmCG`, `z7XH`).
- Efficiency and Reproducibility: Reviewers appreciated the multi-scale RUS estimator’s efficiency improvements and the detailed appendices with code and hyperparameter settings (Reviewer `UcoF`).

To better address the reviewers’ concerns, we have added a set of additional experiments, which are available at the **link below** for easy access. **We have also updated the paper’s PDF accordingly.** A summary of the new experiments is provided here:

1. Synthetic temporal RUS experiments evaluating the multi-scale BATCH estimator under varying sample sizes, feature dimensions, encoder capacities, and numbers of Sinkhorn iterations.

2. Impact of inaccurate R/U/S estimation on TIME-MoE performance.

3. Comparison of training time and maximum GPU memory usage across all methods and datasets.

4. Evaluation of TIME-MoE on a large-scale multimodal generation task to demonstrate scalability and applicability in generative settings.

5. Summary of the training setup for each dataset used in the main paper.

6. How class imbalance for datasets with sequence-level labels affect temporal RUS estimation.

# [Anonymous link to additional results.](https://sites.google.com/view/iclr26timemoe/home)

Sincerely,

Authors

---

### Meta-Review · Area_Chair_EDBX · 2025-12-04

**Summary:**

The submission received majorly positive ratings. It proposes a novel Mixture-of-Experts architecture that integrates temporal multimodal interactions into expert routing via an information-theoretic framework. The method demonstrates strong empirical performance across diverse multimodal benchmarks, particularly in clinical and affective computing domains, while improving the interpretability of expert specialization. The strengths include the novel integration of temporal interaction dynamics into MoE routing, comprehensive evaluation across multiple challenging multimodal datasets, and the improved interpretability of expert activation patterns.

Concerns are raised in the review, including the computational scalability and overhead of RUS estimation, the lack of guarantees or validation for the RUS estimator, sensitivity to hyperparameters and label quality, and the limited exploration of higher-order interactions and scalability to LLM-scale MoEs. The authors provided a detailed rebuttal with additional experiments and clarifications, and addressed most concerns effectively. Acceptance is thus suggested.

**Reviewer Concerns:**

The concerns of the estimator guarantees, label dependency, scalability/cost, computational overhead, and hyperparameter sensitivity are addressed in the rebuttal. While issues like LLM integration and higher-order interactions are acknowledged as future work.

**Reviewer Scores:**

I believe Reviewer RmCG will raise his//her rating from 4 to a higher one. All other reviews are positive.

---

### Decision · Program_Chairs · 2026-01-26

Accept (Poster)